# Dueling dynamics of low-angle normal fault rupture with splay faulting and off-fault damage

J. Biemiller [2,3] ✉, A.-A. Gabriel [1,2] ✉ & T. Ulrich[1]

Despite a lack of modern large earthquakes on shallowly dipping normal faults, Holocene $M_w > 7$ low-angle normal fault (LANF; dip<30°) ruptures are preserved paleoseismically and inferred from historical earthquake and tsunami accounts. Even in well-recorded megathrust earthquakes, the effects of non-linear off-fault plasticity and dynamically reactivated splay faults on shallow deformation and surface displacements, and thus hazard, remain elusive. We develop data-constrained 3D dynamic rupture models of the active Mai'iu LANF that highlight how multiple dynamic shallow deformation mechanisms compete during large LANF earthquakes. We show that shallowly-dipping synthetic splays host more coseismic slip and limit shallow LANF rupture more than steeper antithetic splays. Inelastic hanging-wall yielding localizes into subplanar shear bands indicative of newly initiated splay faults, most prominently above LANFs with thick sedimentary basins. Dynamic splay faulting and sediment failure limit shallow LANF rupture, modulating coseismic subsidence patterns, near-shore slip velocities, and the seismic and tsunami hazards posed by LANF earthquakes.

Complex multifault earthquakes are increasingly observed and can involve unexpected slip dynamics with disastrous consequences, such as during the 2016 $M_w$ 7.8 Kaikoura in New Zealand[1], the 2018 $M_w$ 7.5 Palu-Sulawesi in Indonesia[2], and the 2010 $M_w$ 7.2 El-Mayor–Cucapah in Mexico[3]. Yet, the physical processes and conditions aiding or limiting multi-fault rupture and coseismic splay fault slip in different tectonic settings remain largely equivocal.

Specifically, our understanding of both the physical processes controlling shallow splay fault rupture and the hazard implications of such rupture remain incomplete. Coseismic splay fault activity has been documented and modeled most extensively in subduction zones, where splay fault slip and inelastic deformation in the frontal wedge are intensely debated mechanisms that may amplify coseismic seafloor displacements and resulting tsunami heights of megathrust earthquakes like Tohoku-Oki[4–8]. Vitrinite reflectance and thermal biomarker paleoseismology confirm that such splays slip in large earthquakes[9,10].

The governing factors proposed to control the coseismic slip tendency of megathrust splays include the tectonic stress regime in the upper plate, the frictional properties of splay faults as well as sediments and gouges in the subduction interface, seismically transmitted dynamic stresses from the deeper ruptured megathrust, and enhanced shallow coseismic fault weakening due to thermal pressurization of fluids. However, it is not yet clear which effect is strongest or how they interact. Furthermore, recent models question whether splay faults' contribution to tsunamigenesis is minor compared to that of the deeper megathrust[11].

Recent and ancient complex earthquakes have revealed new normal-fault system slip behaviors and demystified others. Observations from the El-Mayor–Cucapah event suggest that normal faulting earthquakes can involve coseismic rupture of low-angle normal-fault (LANF) segments dipping <30°. Whether LANFs remain active and slip seismically at such shallow dips has

[1]Department of Earth & Environmental Sciences, Ludwig Maximilian University of Munich, Munich, Germany. [2]Institute of Geophysics and Planetary Physics, Scripps Institution of Oceanography, University of California San Diego, La Jolla, CA, USA. [3]Present address: United States Geological Survey, Geology, Minerals, Energy and Geophysics Science Center, Portland, OR, USA. ✉e-mail: jbiemiller@ucsd.edu; algabriel@ucsd.edu

long been debated: Anderson-Byerlee fault theory predicts that LANFs should frictionally lock up and be abandoned in favor of new steeply dipping normal faults[12–14]. Nonetheless, recent paleoseismic evidence[15–18] has shown that LANF systems host large earthquakes that rupture multiple segments of interconnected fault networks. Furthermore, although earthquakes of up to $M_w$ 6.8 have been reported on LANFs worldwide[13,14,19–23], multifault LANF ruptures may explain why $M_w > 7.0$ earthquakes with well-resolved LANF nodal planes are absent in the modern instrumental record: seismic waveforms used in moment tensor inversions sample energy from all rupturing fault segments, and

the contribution from LANF slip may be overprinted by simultaneous seismic slip on more steeply dipping faults[18].

Near the Earth's surface, continental LANF faults typically juxtapose strong metamorphic rocks in the footwall against weaker sediments deposited atop the hanging wall[24] (Fig. 1). The upper few km of LANF hanging walls are often dissected by steeply dipping synthetic and antithetic normal faults that may intersect the LANF[15,25,26] (Fig. 1C, D). Over geologic timescales, the upper portion of a LANF can continue to slip as long as it remains sufficiently weak and well-oriented. Synextensional fault rotation may severely misorient a LANF, such that continued extension breaks and slips along new steeper

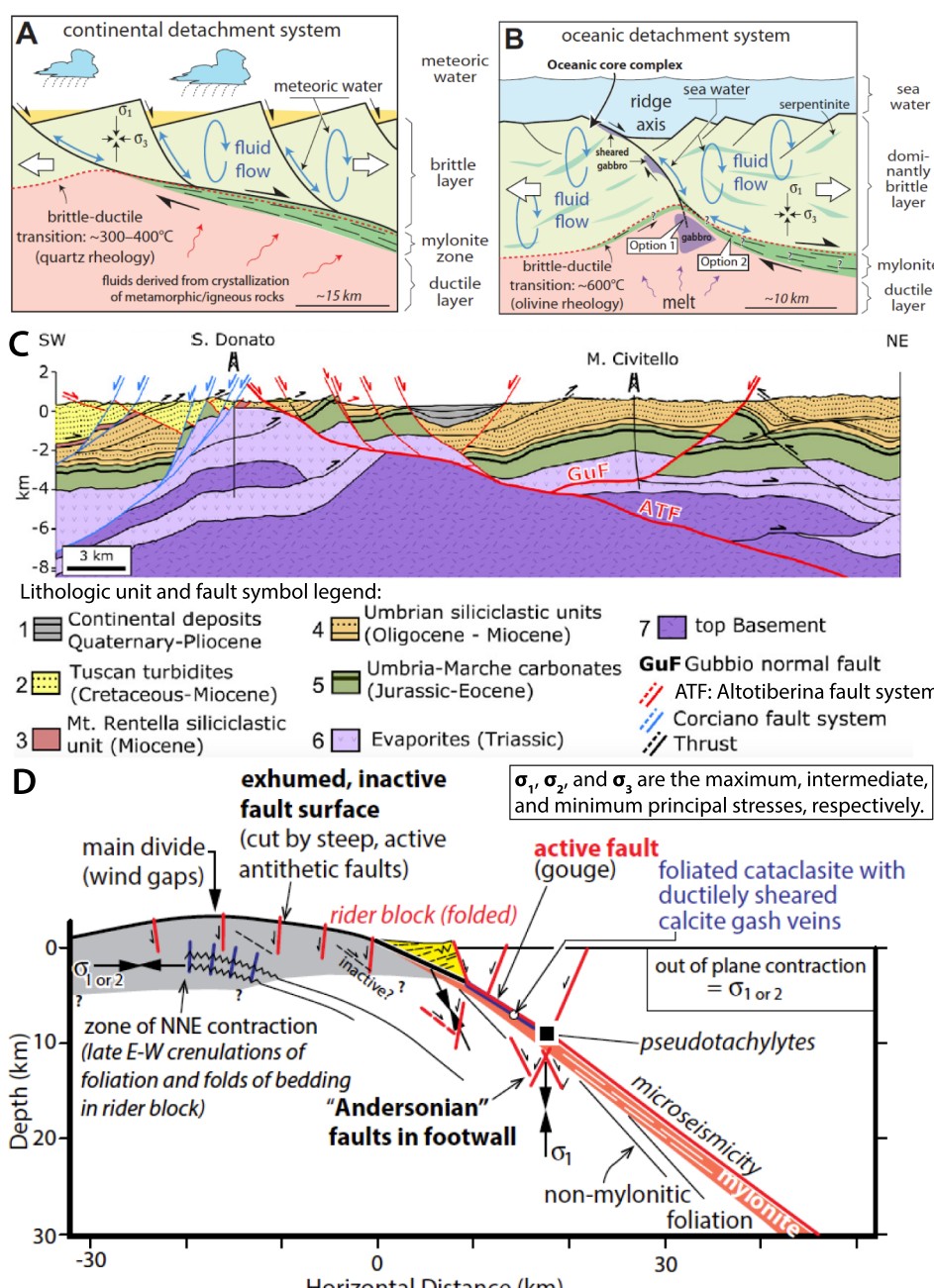

**Fig. 1 | Structure of low-angle normal-fault systems, including subsidiary splay faults. A**, **B** Conceptualization of LANF faulting (modified from ref. 24). Labeled processes highlight how fluid migration, shear zone lithology, crustal structure, and temperature control the mechanical strength and dominant deformation mechanisms of mature continental and oceanic LANF fault systems. **C**, **D** LANF and splay fault architecture of the active Altotiberina (modified from refs. 25,26) and

Mai'iu (modified from ref. 15) fault systems, respectively. Microseismicity, geodetic surface velocities, and scarp geomorphology indicate splay faults in the LANFs' hanging walls are active, but it remains unclear whether these splays slip during large ruptures of their underlying LANFs. See ref. 15 for further details of the inferred paleostress trajectories in the Mai'iu fault footwall.

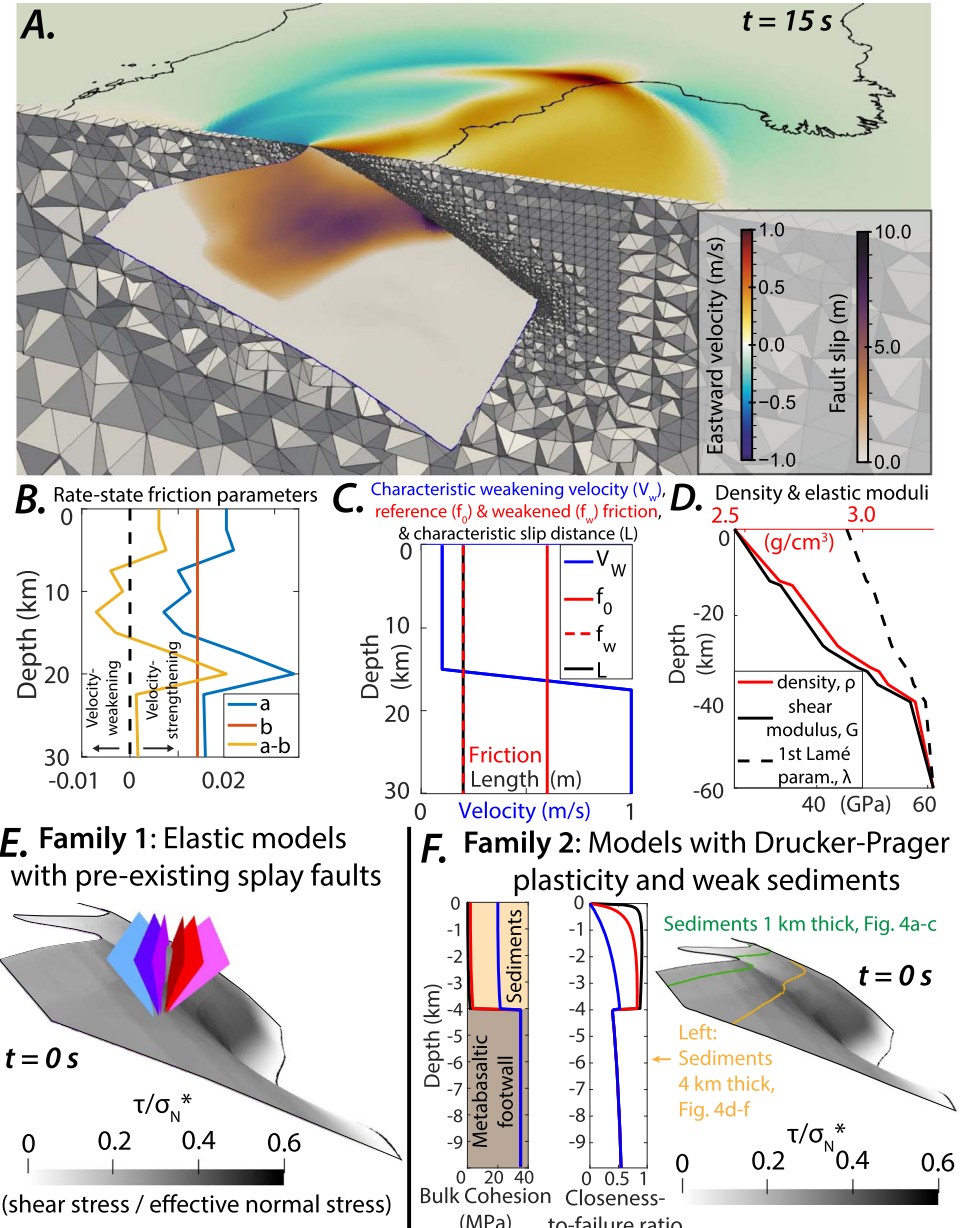

**Fig. 2 | 3D dynamic rupture model setups. A** Example of the geometry, computational mesh, and outputs of one of the Mai'iu fault dynamic rupture models. Plotted snapshot shows cumulative fault slip and instantaneous surface velocities resulting from the model setup of Fig. 4F at the time ($t$) = 15 s. **B–D** Modeled frictional and mechanical properties. Rate-and-state friction parameters are derived from velocity-stepping laboratory friction experiments on exhumed Mai'iu fault materials[76]. Density and rigidity are constrained by seismic velocities beneath the Papuan Peninsula inferred from regional seismic experiments[74,75]. **E** Family 1: Six models with pre-existing splay faults but no off-fault plasticity. **F** Family 2: Six models with non-linear off-fault plasticity, showing plastic cohesion & closeness-to-failure (CF; Eq. (S2)) ratio, which reflects whether materials are close (CF ~ 1) or far (CF ~ 0) from plastic yielding based on the initial stresses (see "Methods"). Note that although E & F show only the upper 15 km of the model, the full modeled fault geometry is identical to that of ref. 31 (Fig. 3), which was derived from ref. 60. Given that ref. 60's surface is largely constrained by onshore mapping along the Dayman–Gwoira segments and microseismicity below 15 km depth, the modeled fault geometry is less tightly constrained and has larger uncertainties towards the along-strike edges and deeper portions of the fault. Supplementary Fig. S10 further illustrates the adapted fault geometries used in our simulations.

splay faults in its hanging wall[27]. This partial or total abandonment of the shallowest LANF can form rider blocks, slices of original hanging-wall material that are subsequently transported atop the footwall[28] (Fig. 1D). The long-term mechanical viability of the shallowest LANF can be conceptualized in Mohr–Coulomb[29] or Mohr–Griffith[30] frameworks as the competition between frictional failure of the LANF or the surrounding crust, such that abandonment depends on the LANF geometry and the relative frictional and cohesive strengths of the LANF and the surrounding crust. Although these studies establish a robust view of the static mechanical processes governing long-term

slip partitioning between a LANF and its splay faults, many questions remain about the coseismic slip behavior, dynamic and static stress transfer, and fault interactions in active LANF systems: Do splay faults initiate during the interseismic period of the LANF due to interseismically elevated stresses, or do they initiate during large LANF ruptures in response to temporarily elevated dynamic stresses? Are existing splay faults coseismically reactivated, or does rupture preferentially propagate to shallow depths along the original LANF?

The active, mega-corrugated, and predominantly concave-down Mai'iu fault in Papua New Guinea (Fig. 2A) dips 16–24° at the surface,

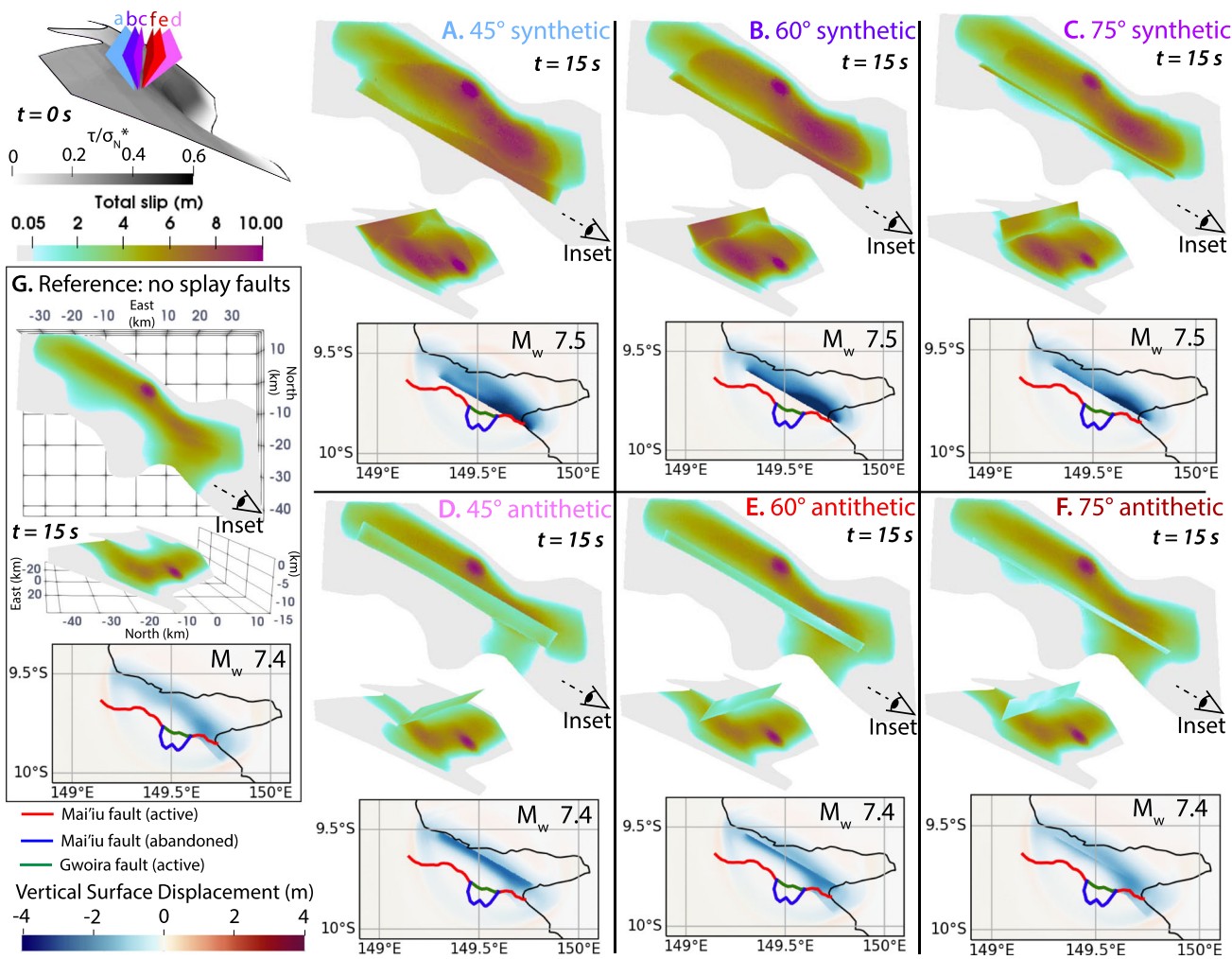

**Fig. 3 | Modeled splay fault geometry modulates competition between coseismic reactivation of pre-existing splay faults and shallow rupture of the underlying LANF. A–F** Fault slip above 15 km depth (top: map view; inset: oblique view along-strike) and surface uplift after 15 s in models with pre-existing splay faults of different geometries. Calculated moment magnitude $M_w$ is shown for each model. Along-strike view in the upper-left panel illustrates the individual modeled splay fault geometries relative to the LANF, along with the initial ratio of the magnitudes of shear stress ($\tau$) to effective normal stress ($\sigma_N^*$) on the LANF. Relative to their antithetic counterparts (**D–F**), synthetic splay faults (**A–C**) host more slip, more strongly enhance and localize surface subsidence, and more efficiently reduce shallow LANF slip. **G** Reference model without splay faults.

exhumes the topographically prominent Dayman–Suckling metamorphic core complex, accommodates horizontal extension of 8 mm/yr, remains interseismically locked from ~5 to 15 km depth, and has hosted both ancient and Holocene earthquakes with inferred $M_w > 7.0$[31,32,15,33,34,17]. Geologically and geophysically constrained dynamic rupture models[31] showed that the Mai'iu fault can host $M_w > 7.0$ earthquakes under Andersonian extensional loading conditions, but that stabilizing effects of the shallowly dipping fault geometry, clay-rich velocity-strengthening gouges, and dynamic stress interactions with the free surface act to inhibit slip near the surface; however, those models only considered the main LANF.

In this work, we assess the effects of coseismic splay faulting and off-fault deformation by performing and analyzing new data-constrained physics-based 3D dynamic rupture simulations of the Mai'iu fault. We incorporate pre-existing splay faults with different geometries and account for dynamic plastic failure in models with overlying sedimentary basins of variable thickness and strength. We find that both splay fault slip and inelastic off-fault damage compete with shallow LANF rupture during large earthquakes; these mechanisms redirect shallow deformation away from the LANF trace, localize and enhance hanging-wall subsidence, and reduce or prevent

slip on the shallowest portion of the LANF. These processes and their effects on shallow coseismic deformation are strongest in the presence of shallowly dipping synthetic splay faults and thicker, weaker sediments.

## Results

### Coseismic reactivation of pre-existing splay faults

We first consider six models subject to Andersonian extension similar to the preferred LANF-only fully elastic earthquake model of ref. 31 (Fig. 2A; see Supplementary Text S1 for details), but with a wider nucleation region combined with stronger overstress and lower, near-hydrostatic pore fluid pressure to balance the effects of off-fault plasticity while yielding comparable earthquake scenarios. The adapted reference model with a larger maximum slip at depth near the hypocenter (7 m) results in an overall slightly larger moment release ($M_w$ 7.4). Each model includes a planar splay fault (Family 1, Fig. 2E) striking parallel to the Mai'iu fault and intersecting it at 5 km depth, dipping synthetically (NNE) or antithetically (SSW) at dip angles of 45°, 60°, or 75° (Fig. 3A–F). For reference, without splays or plasticity, the fault ruptures in a $M_w$ 7.4 earthquake with up to 7 m of slip concentrated between 6 and 13 km depth and limited surface rupture

along only the steeper eastern segment with <4 m of shallow slip (Fig. 3G).

In all models with splay faults, shallow rupture reactivates the splay, reducing or eliminating near-surface slip on the LANF. Partitioning of shallow slip onto the splay faults shifts coseismic subsidence outboard of the fault trace towards the hanging wall (NNE), enhancing and localizing peak subsidence in the immediate hanging wall of the splay fault (Fig. 3A–E). Most notably, the preferential propagation of shallow rupture onto the splay fault is stronger for synthetic splays than antithetic ones and occurs most prominently in the model with the most shallowly dipping synthetic splay (Fig. 3A). Rupture to the surface on synthetic splays generates a strong back-propagating free-surface-reflected rupture front[32] that drives a secondary phase of slip on the deeper LANF (Supplementary Figs. S7 and S8), resulting in higher total slip and larger earthquakes (Fig. 3A–C). In contrast, LANF rupture propagates to the surface past the antithetic splays, inhibiting further splay slip by increasing normal stresses that clamp the splays.

## Off-fault damage in low-angle normal-fault earthquakes

Next, we investigate the effects of coseismic off-fault damage in LANF earthquakes by incorporating non-associated Drucker–Prager plasticity[33] into six dynamic rupture models of the Mai'iu fault. In this framework, plastic failure without dilatancy occurs when stresses locally exceed the plastic yield strength of the bulk rock, determined by the material's internal friction coefficient and bulk cohesion. Laboratory experiments can measure these properties for specific lithologies[34] and Mohr–Coulomb failure analysis of active fault geometries can constrain the relative frictional and cohesive strengths of faults and unfaulted host rock in certain extensional settings[29], but mapping the spatially heterogeneous strength of any region remains challenging. The primary source of crustal strength heterogeneity around the Mai'iu fault is the contrast between the strong metabasalts and ultramafics in its footwall and weaker Plio-Quaternary sediments in the hanging wall[15,35]. In megathrust earthquakes, the amplification of seafloor displacements due to off-fault inelasticity is thought to depend strongly on the thickness and weakness of the shallowest sediments[7,36]. Thus, we analyze six scenarios with thin and thick sedimentary covers (1 and 4 km thick, respectively) modeled by three different frictional and cohesive strength profiles: one with weak unconsolidated sediments, one with partially consolidated intermediated-strength sediments, and one with stronger fully consolidated sediments (Family 2, Fig. 2F and Fig. 4).

With weak and intermediate sediments, dynamic stresses ahead of the updip-propagating rupture front drive localized inelastic off-fault damage in the shallow hanging-wall wedge above the modeled sediment bedrock interface (Fig. 4A, B, D, E). Although damage occurs throughout this wedge, yielding is highly localized along a subplanar plastic shear band dipping synthetic to and more steeply than the underlying LANF. Plastic deformation consumes updip-directed rupture energy and severely limits shallow LANF slip. In contrast, strong sediments exhibit only minor plastic yielding and rupture propagates to the surface relatively unimpeded (Fig. 4C, F). Effects of plastic failure are more pronounced in models with thicker, weaker sediments. For example, the repartitioning of shallow deformation into off-fault damage in the hanging wall reduces coseismic surface subsidence near the fault trace, but strongly enhances subsidence in a localized region above the most severely damaged portion of the hanging wall (Fig. 4D, E, G). In models with pre-existing splays and off-fault plasticity, splay fault slip arrests at the base of the sedimentary basin, above which point plastic hanging-wall deformation dominates (Fig. 4H, I). All models generate slip equivalent to a $M_w$ 7.4 earthquake, though incorporating the equivalent moment due to off-fault plastic damage ($M_P$; Eq. (S4)) increases the total moment magnitude to $M_w$ 7.5 for the model with weak, thick sediments (Fig. 4D). The plastic damage proportion of the total moment ranges from 1.2% for thin, strong sediments to 18.7% for thick, weak sediments (Fig. 4A–F). These ratios are consistent with 2D strike-slip and 3D megathrust dynamic rupture simulations[36] but modest relative to those estimated geodetically from some natural strike-slip ruptures, although the relative contribution of inelastic off-fault damage likely varies widely between faults and is thought to depend strongly on fault zone maturity and width[37].

As in simpler models[31], coseismic footwall uplift is notably minimal (<15 cm), seemingly at odds with longer-term patterns of footwall uplift documented in many LANF systems[24] and recorded along the Mai'iu fault by exhumed fault rocks atop the ~3-km-tall Dayman–Suckling core complex[38] and fossilized coral reefs emerged to >300 m elevation along the triangular-faceted coastline of Goodenough Bay[39,40]. If LANF ruptures involve pronounced hanging-wall subsidence with only minor footwall uplift, as observed in more steeply dipping normal-fault earthquakes[41,42], then protracted LANF footwall uplift must accrue predominantly during the interseismic and/or postseismic periods, possibly via fault-related processes like interseismic creep, afterslip or viscoelastic relaxation, which may occur asymmetrically following large dip-slip earthquakes[43]. Alternatively, broader geodynamic forcings insensitive to local fault locking could drive gradual regional uplift across both the footwall and hanging wall, upon which punctuated upper-crustal LANF earthquakes with minimal coseismic footwall uplift are superimposed, summing to a long-term net vertical displacement pattern with large footwall uplift and minor hanging-wall subsidence. Possible drivers of regional uplift in the highly extended settings where LANFs are commonly found include larger-scale geodynamic processes linked to long-lived localized rifting, such as isostatic compensation of warm, positively buoyant asthenospheric mantle material flowing into regions of thinned mantle lithosphere[44].

## Model uncertainty and comparison with observations

Modeled coseismic hanging-wall subsidence generally matches paleoseismic subsidence estimated from rapidly emerged fossilized coral platforms above the Mai'iu fault (<2 m; Fig. 4G–I). Given the uncertainty in the strike-perpendicular distance between the emerged corals and the active fault trace along the submerged Goodenough Bay segment, all models with plasticity can reasonably match the paleoseismic subsidence of ~1.2 m (Fig. 4G), but those with thick, weak sediments predict greater peak hanging-wall subsidence of up to 5 m further offshore from the emerged coastline. Geologically or historically recorded tsunamis and increased spatial coverage of regional paleoseismic observations would offer additional constraints for validating dynamic rupture models of events without strong modern instrumental records, as demonstrated by paleoseismically constrained partial and full-margin dynamic rupture scenarios of the Cascadia megathrust[45]. At least until a large modern LANF earthquake occurs, data-driven validation and calibration of dynamic rupture models based on paleoseismic and historical datasets from past ruptures allow probing the viability of competing hypotheses of fault system mechanics and dynamics.

Nonetheless, we acknowledge that compared to dynamic rupture modeling studies of well-documented earthquakes, our models are limited by the absence of seismological data from a modern analog earthquake, leading to fewer constraints on and higher uncertainties in instrumentally observable rupture characteristics like hypocentral location, stress drop, and rupture velocity. Furthermore, dynamically simulating paleoseismic ruptures involves facing the same uncertainties and modeling challenges inherent to all dynamic rupture models. For example, although the sensitivity of dynamic rupture to nucleation characteristics has been extensively studied[46–52], observational constraints on earthquake initiation processes remain elusive and the conditions imposed to nucleate ruptures remain some of the less well-constrained components of dynamic rupture models. Nucleation in dynamic rupture simulations can be achieved in a variety of ways, such

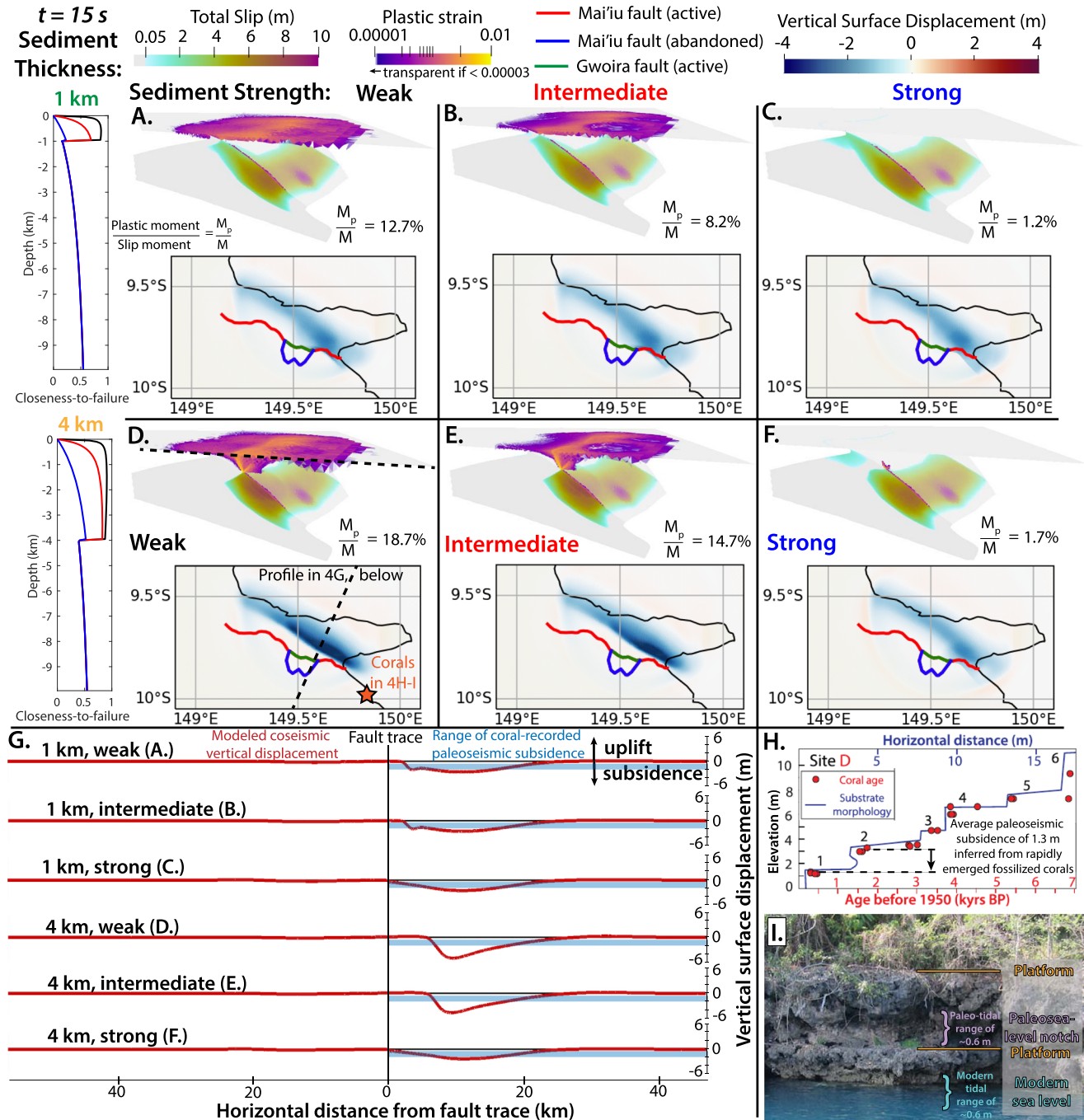

**Fig. 4 | Sediment thickness and strength control modeled coseismic off-fault inelastic deformation and subsidence.** A–F Fault slip above 15 km depth, clipped plastic strain (top: map view; inset: oblique view along-strike), and surface uplift after 15 seconds in models accounting for off-fault inelastic failure with shallow sedimentary basins of variable strength and thickness. Left panels show initial closeness-to-failure for shallow (1 km; **A**–**C**) and deep (4 km; **D**–**F**) basins filled with weak (black), intermediate-strength (red), or strong (blue) sediments. All models with weak and intermediate sediments host distributed damage, while localized damage and subsidence occur only in those with thicker sediments in deeper basins (**D**, **E**). **G** Strike-perpendicular vertical surface displacement profiles from **A**–**F** (red; see profile orientation and location in **D**) relative to paleoseismic subsidence estimated from rapidly emerged coral reefs above the Mai'iu fault (blue; see coral location in **D**). **H**, **I** Morphology and U/Th ages (**H**) of episodically emerged coral platform-notch sequences on the coast of Goodenough Bay, shown in the field photo with sequential sea-level labels in **I** (modified from ref. 17).

as by temporarily reducing fault friction near the hypocenter[53] or temporarily increasing on-fault shear stresses near the hypocenter[31,54]. Beyond implementing one of these methodologies, various strategies exist for assigning values for parameters of the nucleation process such as magnitude, duration, and spatial extent.

Facing few observational constraints on nucleation apart from the pseudotachylite-derived depth of ~10–12 km[15], ref. 31 found and implemented the smallest overstress magnitudes and nucleation radii

that generated self-sustained spontaneous dynamic ruptures in each model configuration, as done in previous studies such as ref. 54 and illustrated in the Supporting Information section of ref. 31. A benefit of this approach is that it minimizes the influence of the potentially ill-constrained nucleation process on the subsequent stages of the simulation. However, implicit to this nucleation procedure is an assumption that interseismic elastic shear stresses should build up in an orderly manner that promotes "minimal" ruptures, accumulating in

a localized region around the optimal hypocentral region and never exceeding the minimum stress concentration needed to nucleate the smallest possible rupture the fault is capable of hosting. In contrast, natural faults are heterogeneous in material, strength, geometry, stress and slip history: thus, it is possible or even common for faults to accumulate heterogeneous sub-critical-to-critical interseismic stresses and strains that eventually contribute to larger-than-minimal earthquake ruptures (e.g., ref. 3; and as illustrated in the creep-rate-derived initial stresses in the Rodgers Creek–Hayward–Calaveras Faults dynamic rupture model setups of ref. 53). While purely elastic single-fault models[31] examined the minimal dynamic ruptures that could have generated coral-recorded 1-m-scale episodic coastal displacements, our new models (Figs. 2–4) show how such offsets could reflect larger events that dissipate rupture energy via a broader array of shallow deformation mechanisms including splay fault slip and off-fault damage. Finally, we note that the systematically dip-variable planar splay faults in our models are broadly representative of the most demonstrably seismically active splay faults in LANF systems like those above the Altotiberina fault[55–59], but that future work incorporating the geometries of mapped and seismically imaged splays of the Mai'iu fault[15,38,60–62] into dynamic rupture simulations could generate more targeted models showing how specific splays might be expected to slip in various rupture scenarios of the Mai'iu fault.

Despite the limitations of modeling uncertainties, our results demonstrate how future data-constrained dynamic rupture models can be constructed for other active LANFs with documented paleotsunamis or denser paleoseismic records, like the Banda LANF[16] or Altotiberina fault[63], respectively. For example, coupling dynamic rupture model displacements to tsunami models[64,65] would allow heights and timing of modeled tsunamis resulting from Banda LANF ruptures to be verified against historical accounts of the 1852 tsunami documented at coastal sites across the Banda Sea[16].

## Discussion

Dynamic rupture models of the Mai'iu fault illustrate how shallow coseismic deformation is accommodated through competing dynamic processes, many of which impede slip on the shallowest part of the LANF. Splay faults in the hanging wall can partially or completely redirect shallow rupture away from the LANF and onto the splays (Fig. 3). While both synthetic and antithetic faults are common above active LANFs[15,25], our dynamic rupture simulations show that shallowly dipping synthetic splays host more coseismic slip and more efficiently limit shallow LANF rupture than their antithetic or more steeply dipping counterparts, likely due to their position and orientation relative to the updip-propagating normal-sense rupture. Our findings agree with generic 2D analysis of mode II shear fractures: ref. 66 showed that when the maximum principal stress is oriented at a large angle to a fault plane, rupture branching occurs most easily for branch faults in the extensional quadrant of the initial rupture; and ref. 67 demonstrated that rupture onto branched faults occurs most efficiently for branches oriented at small angles from the main fault. Our synthetic splay faults are in the extensional quadrant of normal-sense LANF ruptures, and shallowly dipping splays are aligned most closely with the LANF. Furthermore, synthetic splays are better-oriented for slip than the shallow detachment in both the static and dynamic sense: steeper normal faults embedded in Byerlee materials and dipping up to 65–70° are preferentially oriented for slip under Andersonian extensional stresses[12,14,68], while free-surface stress interactions with an updip-propagating normal-fault rupture promotes further rupture propagation on faults dipping 30–75° but inhibits it for those dipping <30°[69]. Despite their favorable orientations, critically stressed splay faults appear unlikely to nucleate large ruptures, slipping instead via bursts of microseismic creep and small earthquakes[57–59] during the interseismic period of the underlying LANF, which may act as a 'keystone fault' that prevents splays from slipping unstably between large LANF ruptures[3].

We note that although all faults are dynamically weak at high slip rates in these models, we do not account for static frictional weakness of mature gouges in the LANF (see model 4 of ref. 31), which could promote slip on the LANF over younger splays with stronger immature gouges. This strength contrast may enhance LANF slip and reduce splay fault slip relative to the modeled slip distributions in Fig. 2. However, features of our modeled ruptures suggest that splay fault slip initiates in response to structurally modulated interactions between dynamic stresses and preferentially oriented pre-existing structures with little regard for shallow fault friction. For example, in the model with a 60°-dipping synthetic splay fault (Fig. 2B), dynamic stresses ahead of the deeper rupture clamp the LANF and unclamp the splay, dynamically initiating spontaneous rupture of the splay fault before the main rupture front reached the LANF-splay intersection (Supplementary Fig. S6). Thus, the effects such static frictional strength variations would have on shallow slip partitioning appear minor relative to first-order structural controls on shallow dynamic slip viability such as those exerted by splay fault geometry. Additional models analyzing variable friction and stress parameterizations based on different degrees of fault maturity (Supplementary Fig. S11) support this conclusion.

Accounting for off-fault plastic yielding (Figs. 3 and 4), we show that localized off-fault damage in weak hanging-wall sediments is an important coseismic process accommodating shallow strain, localizing surface subsidence, and limiting shallow LANF rupture. More extensive yielding in models with thicker sedimentary covers further enhances localized hanging-wall subsidence, suggesting that hanging-wall sediment thickness partially controls whether rupture propagates to the surface along pre-existing faults or arrests at depth after triggering distributed inelastic damage in the shallow hanging wall. While minor plastic yielding distributed through the shallow wedge resembles networks of horsetail splays or branching fractures observed in strike-slip settings[70], the localized planar plastic shear bands that emerge (Figs. 4D, E and 5A, B) resemble discrete coseismically initiated splay faults. Emergent splay fault initiation in sediments not initially stressed close to failure (Fig. 4E) is intriguing, as it suggests that dynamically elevated stresses can break splay faults well before a LANF approaches the static mechanical criteria for partial abandonment and rider block formation[29]. This process may help explain the nature of active synthetic splay faults that have not (yet) captured rider blocks, like those above the Altotiberina fault in the Northern Apennines[25]. Slipping or breaking synthetic splays may be dynamically favorable, while frictionally and cohesively weak clay-rich gouges in the mature LANF keep it weak enough to remain viable for interseismic creep and/or afterslip. Preuss et al.[71] proposed that coseismically initiated faults may form at higher angles to the maximum principal regional stress due to fault-local interseismic stress rotations. Although spontaneously formed shear bands in our models appear well-oriented to regional tectonic loading, these proposed interseismic stress rotations could imply that dynamic splay fault initiation can actually prolong the lifespan of the shallow LANF by breaking the hanging wall along more shallowly dipping splay faults that are less well-oriented for slip than the steeper Andersonian faults expected from static Mohr–Coulomb failure.

Modeled coseismic subsidence is larger, more localized, and farther outboard when the hanging wall is cut by synthetic splay faults (Fig. 3A–C) and overlain by thick sedimentary sections (Fig. 4D–F), as is common above continental LANFs like the Altotiberina. In contrast, more distributed subsidence and limited shallow deformation occur in models without faulted sedimentary hanging walls, like oceanic LANFs near mid-ocean ridges[28]. Localized and enhanced subsidence due to splay fault slip or plastic deformation may increase the tsunamigenic potential of submerged active LANFs like those proposed in the Gulf of

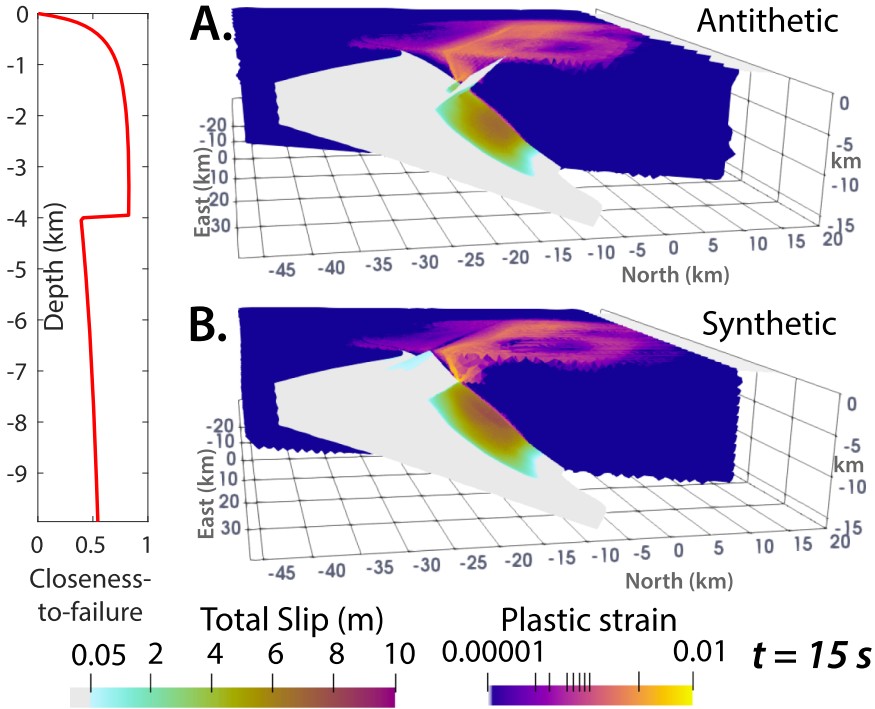

**Fig. 5 | Models with pre-existing splay faults and off-fault plasticity. A, B** Fault slip above 15 km depth and clipped plastic strain after 15 s in models with plasticity, thick intermediate-strength sediments, and pre-existing splay faults dipping 45° antithetic (**A**) or synthetic (**B**) to the LANF, highlighting the dynamic competition between shallow LANF slip, splay fault slip, and off-fault plastic failure. Although shallow deformation is dominated by LANF slip (Fig. 3G) and/or splay fault slip (Fig. 3A, D) in similar models without plasticity, localized and distributed damage outpaces shallow slip when weak sediments and plastic failure are accounted for.

Corinth or Banda Sea. We conclude that the competition between coseismic splay fault rupture, distributed off-fault damage, and shallow LANF slip may be a common feature of large LANF earthquakes that should be considered in seismic hazard assessments of active LANFs.

## Methods

We here summarize key components of our 3D dynamic rupture model setups and methods. Dynamic rupture simulations were performed with the open-source software SeisSol (www.seissol.org), which can solve for simultaneous seismic wave propagation, frictional dynamic earthquake rupture, and off-fault plastic deformation. All models solve for frictional failure and fault slip governed by rate-and-state friction with enhanced velocity-weakening at fast slip rates[72] which accounts for enhanced weakening observed in laboratory experiments at such slip rates[73]. Models in family 2 (Figs. 2E and 4) additionally solve for non-linear non-associative Drucker–Prager plastic failure[33,36] in the model volume off-fault; models in family 1 (Figs. 2d and 3) do not.

### Mechanical properties and regional lithospheric strength

Geologic and geophysical observations from the Mai'iu fault constrain dynamic rupture model parameters including the non-planar fault geometry, initial stress orientations and magnitudes, and depth-dependent distributions of elastic moduli, plastic strength parameters, and fault frictional stability determined by rate-and-state friction parameters[31], which are implemented as shown in Fig. 2. Heterogeneous density and elastic moduli (Fig. 2D) are derived from the Papuan Peninsula regional seismic velocity models of refs. 74,75. Frictional stability (Fig. 2B, C) is based on velocity-stepping friction experiments on exhumed Mai'iu fault materials[76], which show transitions from velocity-strengthening to velocity-weakening and back to velocity-strengthening with increasing depth, temperature, and confining stress.

### Structure and stress

The geometry of the Mai'iu fault is constrained by field mapping and tectonic geomorphologic analysis of the exhumed footwall[15,38,60], along with microseismicity that delineates the downdip extent of the fault[75]. Geodetically and geomorphically confirmed strike-perpendicular horizontal extension[38,76–78] and active synexhumational folding and corrugation of the fault surface with strike-parallel shortening[15,60] constrain the orientations of ratios of the principal stresses: the maximum principal stress ($\sigma_1$) is vertical; the intermediate principal stress ($\sigma_2$) is horizontal along-strike (N60W); the least principal stress ($\sigma_3$) is horizontal and aligned parallel to extension (N30E); and the ratio of principal stress magnitudes $\Phi = \frac{\sigma_2 - \sigma_3}{\sigma_1 - \sigma_3}$ is 0.8 as constrained by stress inversion of mapped minor faults and paleopiezometric orientations from syntectonic calcite veins in the exhumed footwall[79].

Above 15 km, stresses are computed for Andersonian extension (vertical maximum principal stress) with optimally oriented segments initially stressed close to failure. The latter is achieved by computing the local relative prestress ratio $R$ relative to the maximum relative prestress ratio $R_O$, with $R = R_O$ for optimally oriented fault segments and $R < R_O$ for all others. $R$ is defined as:

$$R = \frac{\Delta\tau}{\Delta\tau_b} = \frac{\tau_0 - \mu_d\sigma_n'}{(\mu_s - \mu_d)\sigma_n'} \approx \frac{\tau_0 - f_w\sigma_n'}{(f_0 - f_w)\sigma_n'} \tag{S1}$$

based on the initial shear stress $\tau_0$, the initial effective normal stress $\sigma_n'$, the estimated equivalent static friction coefficient $\mu_s \approx f_0 = 0.6$, and the estimated equivalent dynamic friction coefficient $\mu_d \approx f_w = 0.2$. Possible values of $R_O$ range from <0 (minimal stress on optimally oriented segments) up to 1 (critically stressed optimally oriented segments). Here, we set $R_O = 0.95$. Local fault orientation controls the resulting stress magnitudes following the approach of ref. 80 with normal-sense slip according to $\Phi = 0.8$, near-hydrostatic pore fluid pressure ratio $\lambda_f = P_{fl}/\rho gz = 0.44$, $R_O = 0.95$, the reference friction coefficient $f_0 = 0.6$, and the azimuth of $\sigma_2$ (N60W). A smooth

stress taper[80] reduces the resolved deviatoric stresses from 11 to 15 km in accordance with the microstructurally and paleopiezometrically recorded brittle–ductile transition zone of the Mai'iu fault rocks[15,79,81].

## Earthquake nucleation and peak slip rates

Earthquake nucleation is achieved by applying smoothly over time a Gaussian pattern of increased shear tractions. Off-fault plastic deformation consumes a portion of available rupture energy, and thus, stronger nucleation tractions are required to generate self-sustained rupture when accounting for off-fault plasticity than in models without it[52,82]. In addition, the lower pore pressures in our models necessitate larger overstresses to initiate sustained dynamic rupture than those used in the purely elastic single-fault models of ref. 31. Here, we impose a stronger nucleation than in ref. 31, consisting of a maximum traction perturbation of $T_{nuc} = 45$ MPa over a spherical region of radius $r_{nuc} = 3$ km centered on the fault at 11 km depth for a duration of $t_{nuc} = 1$ s. The effects of different nucleation conditions and hypocentral locations on single-fault LANF ruptures are detailed in ref. 31.

Self-sustained rupture can dynamically outpace the various mechanisms that impede shallow slip on low-angle normal faults. When combined with the experimentally-derived strongly weakening behavior of the cataclastic Mai'iu fault rocks[76] in the enhanced coseismic weakening rate-and-state friction law we use[72], our models generate high peak slip rates locally exceeding 10 m/s (Supplementary Fig. S1). Although parameters in our dynamic rupture models are constrained from regionally specific field and laboratory observations wherever possible, inherent uncertainties in the measurement and extrapolation of physical parameters are unavoidable and suggest that further study of rupture processes will continue to improve our parameterizations of such models. One parameter with limited observational constraint is the characteristic slip distance, $L$, which determines the slip distance over which the rate-and-state frictional response to a change in slip velocity occurs (specifically for slip rates lower than $V_w$, above which the enhanced velocity-weakening becomes the dominant weakening mechanism). Selecting the appropriate values of $L$ for dynamic rupture simulations is challenging, given discrepancies in estimates of the critical value of $L$ for unstable seismic slip (the critical slip distance) from different methods[83]: laboratory measurements indicate this distance is on the order of $10^{-5}$ m, while analytical frictional stability modeling suggests $L$ in natural earthquakes could be as low as $10^{-2}$ m. In addition, $L$ for a single fault may vary laterally along-strike or with depth.

We find that in our dynamic LANF rupture models, the characteristic slip distance $L$ exerts a first-order control on peak slip rates by modulating the intensity and timing of the onset of velocity-weakening in response to a variation in slip rate. In Supplementary Fig. S1, we show the peak slip rates arising from models with different values of $L$. In models with spatially homogeneous $L$, increased $L$ leads to decreased peak slip rates, delayed nucleation, and eventually failed nucleation with large enough $L$ (~1 m). Heterogeneous models with smaller $L$ near the hypocenter and larger $L$ away from the hypocenter reduce peak slip rates without stalling rupture nucleation for a wider range of nucleation conditions. In addition, Supplementary Fig. S5 shows that different values of $L$ and corresponding peak slip rates may influence patterns of coseismic off-fault deformation, with larger $L$ and lower peak slip rates resulting in deeper-seated subplanar plastic deformation bands (interpreted as incipient splay faulting) than models with smaller $L$ and lower peak slip rates. Future modeling efforts which may target non-ergodic, physics-based strong ground motion modeling should further explore the sensitivity of peak slip rates to model parameters, including the plastic yield strength controlling off-fault inelastic deformation[82].

## Off-fault plasticity and bulk rock strength

Off-fault inelastic deformation is accounted for using a Drucker–Prager elastoviscoplastic rheology[33,36]. Depth-dependent distributions of bulk friction coefficient, $v$, and plastic cohesion, $C$, approximate the crustal strength contrast between weak, variably consolidated sediments deposited in the hanging-wall basin and stronger metabasaltic lithologies in the footwall[15]. Below the modeled sediment-basement interface, the strong metabasalts are assigned a bulk friction coefficient of 0.85 and a bulk cohesion of 35 MPa. Above this interface, weaker sediments have a bulk friction coefficient of 0.6 and bulk cohesions of 0.2 MPa, 2 MPa, or 20 MPa. The exact distributions of shallow plastic cohesions are shown in Fig. 2F. These variable cohesive strengths of the modeled sediments lead to hanging walls that are initially stressed closer or further from plastic failure than in other models. How close these sediments initially are to plastic failure can be quantified by the closeness-to-failure ratio[33] $CF$, defined as:

$$CF = \frac{\sqrt{I_2}}{\tau_c} \tag{S2}$$

based on the second invariant of the deviatoric stress tensor, $I_2$, and the Drucker–Prager yield criterion $\tau_c$, given by:

$$\tau_c = C(z)\cos(\Phi) - \sigma_m \cos(\Phi) \tag{S3}$$

with the internal friction angle $\Phi = \arctan(v)$, the mean stress $\sigma_m = \sum_{n=1}^{3} \frac{\sigma_{ii}}{3}$, and the depth-dependent bulk cohesion $C(z)$. CF ranges from 0 (no stress; far from failure) to 1 (critically stressed; at failure) and is plotted for the six different models with plasticity in Fig. 4 to illustrate how close the surrounding crust initially is to being critically stressed in these models. The equivalent moment due to plastic strain ($M_p$; Fig. 4A–F) is calculated from the inelastic strain rate, $\dot{\epsilon}^p_{ij}$, following refs. 36,84:

$$M_p(t) = \int_0^t \sqrt{\frac{1}{2} \dot{\epsilon}^p_{ij} \dot{\epsilon}^p_{ij}} \, dt \tag{S4}$$

## Data availability

No new datasets were acquired as part of this study. All input files required to reproduce the SeisSol models, including the computational meshes, are publicly available via the dedicated Zenodo repository at https://doi.org/10.5281/zenodo.7478660.

## Code availability

SeisSol (www.seissol.org) is an open-source software freely available to download from https://github.com/SeisSol/SeisSol/. We use branch Maiiu/f0_variable (https://github.com/SeisSol/SeisSol/tree/Maiiu/f0_variable) and commit #26c02f2. Instructions for downloading, installing, and running the code are available in the SeisSol documentation at https://seissol.readthedocs.io/. Downloading & compiling instructions: https://seissol.readthedocs.io/en/latest/compiling-seissol.html. Instructions for setting up and running simulations: https://seissol.readthedocs.io/en/latest/configuration.html. Quickstart installations and introductory materials are provided in the docker container and jupyter notebooks at https://github.com/SeisSol/Training. Example problems and model configuration files are provided at https://github.com/SeisSol/Examples, many of which reproduce the SCEC 3D Dynamic Rupture benchmark problems described in refs. 85,86, and at https://strike.scec.org/cvws/benchmark_descriptions.html. Plots in Figs. 2–5 were made with Paraview (https://www.paraview.org/) version 5.5.0-RC3 64-bit. All input files required to reproduce the SeisSol models, including the computational meshes, are publicly available via the dedicated Zenodo repository at https://doi.org/10.5281/zenodo.7478660.

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

## Acknowledgements

We acknowledge support from the Green Foundation (IGPP, SIO, UCSD; J.B.), European Union's Horizon 2020 research and innovation program (TEAR ERC Starting grant agreement No. 852992; A.-A.G.) and Horizon Europe (DT-Geo grant agreement No. 101058129, Geo-Inquire grant agreement No. 101058518, ChEESE-2P grant agreement No. 101093038; A.-A.G.), the Bavarian State Ministry for Science and Art project Geothermal-Alliance Bavaria (GAB), the National Science Foundation (NSF) (EAR-2121568; A.-A.G.) and the Southern California Earthquake Center (SCEC) (21112, 22135; A.-A.G.). The authors acknowledge the Gauss Centre for Supercomputing e.V. (www.gauss-centre.eu, project pr63qo) for funding this project by providing computing time on the GCS Supercomputer SuperMUC-NG at Leibniz Supercomputing Centre (www.lrz.de).

## Author contributions

J.B. and A.-A.G. initiated the study. J.B., A.-A.G., and T.U. designed the experiments. Modeling was conducted by J.B. with guidance from A.-A.G. and T.U. The manuscript was jointly written by J.B., A.-A.G., and T.U.

## Competing interests

The authors declare no competing interests.
