## [Peer Review File · Nature Communications]

REVIEWER COMMENTS

Reviewer #1 (Remarks to the Author):

This is a well written paper focused on investigating the possible effects to dynamic rupture simulation models of an active low-angle normal fault of adding further complications to an already complicated 3D rupture model. The authors investigate the parameter space of adding a more steeply dipping splay of variable dip (both synthetic and antithetic) and the possible effects of shallow off-fault sediment deformation of various strength and thickness in the hanging wall. These results are of some interest to those concerned with how deep fault slip during earthquakes can be potentially manifested at the surface, and especially to those concerned with the degree to which adding elements of increased fault or off-fault complexity can potentially enhance our understanding of how dynamic rupture deformation may be expressed in the shallow geologic record. However, owing to major problems associated with unexplained discrepancies between the previously published reference model (Biemiller et al., 2022) used as the basis for this study, lingering questions regarding model formulation and results, and the tendency for these additional elements of model complication to move the results further away from the limited geologic controls and observations used to infer the first-order characteristics of the rupture model, this tends to severely limit the significance and usefulness of this study, and makes it unpublishable in its current form. Major discrepancies with published reference model as presented in this study

1) Although the color scheme of shaded total slip (m) for the reference model presented in Fig.3g appears identical to the published reference model (Fig.5, Biemiller et al., 2022), there is in fact an unexplained factor of two discrepancy in color bars. Color bars that previously ranged from 0.05 to 5 m now range from 0.05 to 10 m for the same color gradations. This implies that the total slip at depth, that previously ranged from 3-4 m are now 6-7 m for the exact same model even though the resulting magnitude (Mw 7.4) did not change. This is not possible. Although one might suspect that there has been a simple mistake in the color bar assignment for the figure (as it looks like they mistakenly used the color bar for Fig.S1, rather than the correct one from Fig.5), this is apparently not the case. Thus, where previously the total slip at depth was on the order of 3-4 m between 6-13 km depth (Fig.5, Biemiller et al., 2022), the authors now claim that the reference model "ruptures in a Mw 7.4 earthquake with up to 7 m of slip concentrated between 6-13 km depth" (Line 132). The reference model now produces nearly twice as much slip as it did before. This also implies that with the addition of the hanging-wall synthetic splay fault, total slip at the depth increases up to 8-9 m in the same region of the deep detachment. What is the physics of this dramatic doubling of the total slip on the deep detachment? Given that both the slip and fault rupture area have increased significantly owing the presence of a synthetic splay fault, the moment and magnitude must also increase accordingly from Mw 7.4 to Mw 7.8, but the authors claim it is only Mw 7.5.

2) The published reference 3D model does not explicitly document the change in dip along strike or down-dip. The reader might assume that the 2D geometry shown in Fig.1d (Biemiller et al., 2022) might apply for most of the reference model, but this is apparently not the case. Although it is difficult to tell from the available 3D perspective views, the eastern edge of the model tends to show a smooth curved detachment that steepens from near 20° at the surface to about 30° at depth (Fig.3a, Biemiller et al., 2022). The more steeply dipping section used for Goodenough Bay that allows rupture to propagate to shallow depths apparently dips at 35° and this steeper dip extends to depths of 5 km or more. The problem is that, although it is again difficult to tell from the limited 3D perspective views provided, it appears that for the reference model used in this study, there is a change of ~30° in the dip of the shallow detachment at about 2 km depth for the far eastern end of the model from a nearly planar deeper detachment that dips at about 20° to a shallow more steeply dipping segment that dips at about 50° (Fig. 2E & 2F, this study). This change in model geometry relative to the original reference model (Fig.3a, Biemiller et al., 2022) is not explained or even identified.

Lingering questions of model formulation and results

3) From the original formulation of the reference model, total slip at depth is about 3-4

m on a fault that dips at 30° . This produces a maximum vertical displacement (subsidence) of 2 m. However, if this displacement is buried at depths of 6-13 km, it is unlikely to be fully expressed at the surface. Buried or blind ruptures with meters of displacement (e.g., Northridge, Loma Prieta, etc.) rarely produce surface displacements more than a few 10's of centimeters. However, the authors assert that such buried fault slip will yield vertical surface displacements on the order of 1 m or more (Fig.5, Biemiller et al., 2022). Although I am not an expert in modeling geodetic changes resulting from buried fault slip, this still does not seem realistic to me. Yes, if this slip reaches the surface, as it is modeled to do in Goodenough Bay, one might expect up to 1.1 m of related vertical surface displacement, but not farther outboard above where the fault slip is buried at depths of 6 km or more. The authors thus tend to imply that whether fault slip reaches the surface or is buried at depths below 5 km, the vertical surface displacement is the same. This does not make sense to me.

4) For the added complication of a hanging-wall splay fault in this study, although it is certainly interesting to see the results of investigating the parameter space of variable dip and dip direction (from synthetic to antithetic), I was curious as to why the authors did not use the locations and dips of the actual hanging-wall splay faults imaged, mapped and defined at depth with MCS reflection data (e.g., Lines 1190, 1191, 1193, Fitz and Mann, 2013 a,b). These results suggest few if any mapped splays occupy the seafloor outcrop positions proposed for the modeled splay fault, or would even project to intersect the detachment at 5 km depth, as all the proposed modeled splay faults in this study do. There is also the issue that most of the mapped splay faults located farther west on the western side of the detachment (e.g., Line 1181) are inactive and are not imaged to offset Pleistocene and younger sediments.

5) In terms of the dynamic behavior of the more steeply dipping splay faults, the question is whether these faults with little cumulative offset would be similar in terms of static friction, strength and rupture behavior to the detachment with known large cumulative offset, low static friction, and velocity-strengthening materials for shallow fault rocks. It seems that the authors assume the splay has the same characteristics as the detachment at shallow (<5 km) depths. This does not seem likely to me given the steeply dipping splays have significantly less cumulative offset, are younger, and often cut different lithology than the detachment. Thus, although the more steeply dipping splay may be more favorably oriented for normal slip relative to the regional stress field, its other characteristics may be such that slip on the weak low-angle detachment may still be favored to some extent.

6) It is helpful that the first-order model results of adding a hanging-wall splay seem to be consistent with expectations: a) antithetic splays are not particularly favored to localize shallow slip or to alter the slip on detachment, especially for steeper dips; b) on the other hand, synthetic splays tend to localize slip at shallow depths and increase slip on the deeper detachment, likely owing to an increase in dynamic shear stress; c) but increasing dip of the synthetic splay tends to reduce the slip on the deep detachment by increasing the normal (clamping) stress. This is all well and good. Unfortunately, it highlights that adding this additional fault complexity of a hanging-wall splay also tends to move the model results farther away from the limited observations used to define first-order rupture model parameters.

Increased model complexity tends to yield results less consistent with observations

7) The primary geologic control used to define the reference model parameters are the locations and heights of exhumed, offset footwall corals near the coast of Goodenough Bay (Biemiller et al., 2020, 2022). The inferred average vertical surface displacements (~1.2 m) were used to define the length, width, moment and M_w of the simulated rupture. However, if linearly reduced static friction levels extending from 0.6 at 12 km depth to 0.25 at the surface are imposed on the reference model (Model 4, Biemiller et al., 2022), which seem to be more consistent with the observed fault rocks, the simulated earthquake is reduced to M_w 7.0 and rupture does not penetrate to the surface along any segment. Thus, adding somewhat more realistic static friction levels up-dip along the detachment, tends to decrease the ability of the rupture model to match the observations of the offset corals.

This issue that adding model complexity tends to move the results farther away from the limited geologic controls and observations used to define the model is also a problem in

this study that investigates adding the increased complexity of a hanging-wall splay or off-fault sediment deformation. As the authors state: "Modeled coseismic subsidence is larger, more localized, and farther outboard when the hanging wall is cut by synthetic splay faults (Fig. 3A-C) and overlain by thick sedimentary sections (Fig. 4D-F)".

Although the subsidence is larger, the location of surface deformation is much farther offshore where it is not observed, if the exposed, exhumed, coastal offset corals near the outcrop of the detachment are an indication of the principal surface displacement location.

Summary

The major discrepancies in total slip and fault geometry between the published reference model and the version of the same reference model used in this study, the outstanding questions of model formulation and results, the inconsistency between model splay fault geometry and imaged splay faults defined by MCS data, and the tendency for added model complexity of splay faults and/or off-fault deformation to move model results farther away from the limited geologic controls used to define first-order model parameters, severely limit the significance and usefulness of this study, and makes it unpublishable in its current form.

Reviewer #2 (Remarks to the Author):

Review of "Dueling dynamics: competition between detachment rupture, splay faults, & off-fault damage"

By Biemiller et al.

Submitted to Nature Communications

Review by Andrew Meigs

Key Results

This study addresses the impact that (1) fault splays and (2) off-fault damage have on rupture of a low angle normal fault (LANF). Dynamic rupture simulations explore the resultant slip using a model based on an extensively studied, archetypal low angle normal fault in New Guinea. Synthetic fault splays (synthetic = dip in the same direction as the master LANF; antithetic = dip in the opposite direction of the LANF) tend to enable connectivity between the master fault and splays and to promote slip to the surface in the simulations. Antithetic splays impede connectivity and rupture to the surface. Off fault damage is explored by comparing models with a strong footwall and thin and thick hanging wall basin fill of variable sediment strength. Distributed damage is associated with weak and intermediate strength sediment and increase in seismic moment relative to a reference model. Models with thick sediment yield coseismic hanging wall subsidence and up to 18% greater seismic moment than either the reference model or models with strong sediment.

Validity

Whereas I am not a modeler, the method appears robust as it employs published code and inputs, the 2 families of models explore topical and appropriate structural and rheological scenarios and the results are conservatively interpreted. I especially appreciated supplemental figure S2-4, which compared the effect of color scheme on the appearance of model runs.

Significance

This study is significant because it builds on an outstanding body of observations, lab measurements and other data from the Mai'iu fault in New Guinea. The fault is one of the most comprehensively studied low angle normal faults in terms of its structures on all scales, its structural evolution, the rheological characteristics of the exhumed fault zone rocks and numerous other aspects of its geological development. Thus, prior studies provide key constraints on the mechanical and rheological boundary conditions of the modeling. My understanding is that the dynamic rupture modeling utilized in this analysis is state of the art. Because so few low angle normal fault earthquakes have demonstrably ruptured seismically, this study provides synthetic outcomes that forecast potential rupture and deformation scenarios in a future earthquake. That future event will provide the sort of data to both calibrate the model and assess its applicability. The three most striking results from my point of view are (1) that synthetic splay faulting is

a potential prerequisite to surface rupture, (2) that antithetic faults potential represent barriers to slip transfer and earthquake size and (3) that low basin sediment strength likely plays a role in seismic moment, in positive feedback for basin subsidence and new splay fault formation due low strength sediment's potential for enhanced off-fault damage.

Data and methodology

There aren't really any data presented in the study. The one piece of data is a small photo of a coral outcrop and a relative sea level curve derived from age data from that outcrop.

Analytical approach

I am not qualified to assess the analytical approach in detail. The conceptual framework, variables and concepts are consistent with my understanding of dynamic and static friction, low angle normal fault geometrical and rheological considerations, and coseismic slip distributions on faults.

Suggested improvements

1) Restrict your terminology to "low angle normal fault" and do not use the "detachment" synonym. Whereas both terms appear in the literature, low angle normal faulting is what you seek to understand and is a clearer and more specific description.
2) Clarify why you include so much discussion of subduction megathrust models and earthquakes. Are they included because splay faulting is topical for those kinds of events, because dynamic rupture modeling has focused on that setting or for some other reason? In the case of megathrust events, I understand that the role of splay faults in tsunamigenesis is a hot topic and that a low angle normal fault in the marine realm has potential to generate a tsunami. That said, I am unsure of the importance of that outcome, particularly considering that the Mai'iu fault and splays crop out onshore according to the maps in Little et al. (2019).

3) Be more holistic in your reporting of active low angle normal faults and earthquake events on them. The statement on lines 61-62 that focuses on the absence of LANF events > Mw 7, fails to highlight that a number of M6 events are documented in papers such as Axen (1999), Abers et al. (1997), Abbott et al (2001), Bernard et al. (1997) and Wernicke (1995).

Abers, G. A., Mutter, C. Z., and Fang, J., 1997, Shallow dips of normal faults during rapid extension: Earthquakes in the Woodlark-D'Entrecasteaux rift system, Papua New Guinea: *Journal of Geophysical Research: Solid Earth*, v. 102, no. B7, p. 15301-15317.

Abbott, R. E., Louie, J. N., Caskey, S. J., and Pullammanappallil, S., 2001, Geophysical confirmation of low-angle normal slip on the historically active Dixie Valley fault, Nevada: *Journal of Geophysical Research*, v. 106, no. B3, p. 4169-4981.

Axen, G. J., 1999, Low-angle normal fault earthquakes and triggering: *Geophysical Research Letters*, v. 26, no. 24, p. 3693-3696.

Bernard, P., Briole, P., Meyer, B., Lyon-Caen, H., Gomez, J.-M., Tiberi, C., Berge, C., Cattin, R., Hatzfeld, D., and Lachet, C., 1997, The Ms= 6.2, June 15, 1995 Aigion earthquake (Greece): evidence for low angle normal faulting in the Corinth rift: *Journal of Seismology*, v. 1, no. 2, p. 131-150.

4) Provide some comment as to why your models produce no footwall uplift (i.e. figure 3 and 4). Normal fault earthquakes such as the Borah Peak event in 1983 (Stein et al., 1988) and long-term geological extension produce substantial footwall uplift (Davis and Lister, 1988). Data from the footwall block of the Mai'iu fault indicate that footwall uplift is on the order of 5 – 10 km, many times larger than the hanging wall subsidence. A similar imbalance in footwall uplift relative to hanging wall subsidence is seen in the long term geological evolution of most low angle normal fault systems.

Davis, G. A., and Lister, G. S., 1988, Detachment faulting in continental extension; Perspectives from the southwestern U.S. Cordillera, in Clark, S. P., Burchfield, B. C., and Suppe, J., eds., *Processes in Continental Lithospheric Deformation*, Volume 218: Boulder, Geological Society of America, p. 133-160.

Stein, R. S., King, G. C. P., and Rundle, J. B., 1988, The growth of geological structures by repeated earthquakes 2. Field examples of continental dip-slip faults: *Journal of Geophysical Research*, v. 93, no. B11, p. 13,319-313,331.

5) Minor point – I do not understand how the relative sea level fall revealed by the coral data are indicative of coseismic subsidence. In a subduction setting, coseismic

subsidence is indicated by an abrupt sea level rise because the Earth's surface moves down relative to sea level. This is true for coastal estuaries (Atwater and Hemphill-Haley, 1997) and for coral heads above the Sumatra subduction zone (Zachariassen et al., 1999).

Atwater, B. F., and Hemphill-Haley, E., 1997, Recurrence intervals for great earthquakes of the past 3,500 years at northeastern Willapa Bay, Washington: USGS Profession Paper 1576, US Government Information Services [distributor], 92 p.

Zachariassen, J., Sieh, K., Taylor, F. W., Edwards, R. L., and Hantoro, W. S., 1999, Submergence and uplift associated with the giant 1833 Sumatran subduction earthquake; evidence from coral microatolls: *Journal of Geophysical Research*, v. 104, p. 895.

Clarity and context

The manuscript is clearly written. The context well articulated and the scope is reasonable.

References

See above comments for potential additional sources. Otherwise the references are appropriate and comprehensive.

My expertise

The focus of my research and teaching are structural geology, continental tectonics and geodynamics and earthquake geology.

Reviewer #3 (Remarks to the Author):

Review for "Dueling dynamics: competition between detachment rupture, splay faults, and off-fault damage."

By J. Biemiller, A.-A. Gabriel, and T. Ulrich

The topic of this paper is very interesting and at the same time challenging one. The authors study the dynamic interaction between low angle normal faults and splay faults connected to the main structure. To do that they use both elastic and plastic models dynamic rupture models.

Low angle normal faults are by nature "enigmatic" with respect to the classical Andersonian theory for fault activation. Although we do not have any recent examples of major earthquakes where the main rupture occurs on a low angle normal fault, structures such as the Altotiberina fault in Italy could host damaging earthquakes. Same for the Mai'iu Papua New Guinea fault discussed in this paper.

The paper (and corresponding dynamic rupture finite element model) is based on a previously published work investigating the dynamics of shallow ruptures along the main detachment. The new configuration includes six models each one corresponding to a different splay fault model with increasing dipping angles (45 to 75) and synthetic or antithetic geometry.

The family of models with synthetic (45 and 60) splay geometries shows two main and interesting observations when compared to the antithetic counterpart. First, rupture propagates easily over the splay fault, and second, the magnitude of slip is higher in the main detachment. In other words, rupture propagation on the splay fault, promotes higher slip in the main detachment. The antithetic configuration appears to have the opposite effect (act more like a barrier) and show lower slip on the main detachment. In addition to the experiments with purely elastic rheology the authors presented a series of simulations with off-fault plastic failure, that allow them to reproduce observed subsidence detected using corals. The "plastic" experiments showed that when the sediments are weak and thick the model produces localized off-fault deformation matching historical subsidence. As expected, "strong" sediments do not show significant plastic yielding and for that reason generate a weaker and smoother subsidence pattern above the hanging wall.

This is a well written and nicely presented paper and I believe deserves publication after some clarifications/discussion regarding the results presented here. If you don't have space in the main text to provide comments/clarifications, you could extend the supplementary material:

1. In all the simulations presented in this paper the shallowest part of the main fault

interface is velocity strengthening, inhibiting the propagation of rupture near the intersection with the free surface. We can appreciate this effect in the final slip map in the model with "no splay faults" (Figure 3G). Would that effect somehow "isolate" and promote rupture along the splay faults and basically create a preferential direction of rupture? In other words what would happen if rupture were allowed to propagate on both the splay fault and shallow part of the main detachment? I would expect for example that normal motion on the shallow detachment (after rupture passes the intersection with the splay and propagates towards the free surface) could potentially increase normal stress on the splay inhibiting rupture and consequently decrease the amount of final slip. When I first read the title of the paper, I thought that "Dueling" was referring to competing ruptures on the main detachment and the splay fault.

2. Have you looked specifically at the normal stress changes ahead and behind the rupture front and how such changes may affect the outcome of your simulations? Would rupture along a normal fault produce clamping or unclamping ahead of the rupture front? How such process affects the interaction with the splay fault?

3. Does the speed of rupture on the main detachment (slow vs fast rupture) plays any role on how efficiently the system activates the splay faults?

4. Could a rupture start from a splay fault instead of the main detachment?

5. A short discussion on what could be some limiting factors in your work. For example, would a realistic surface topography in your FEM make any difference in your simulations?

REVIEWER COMMENTS

Reviewer #1 (Remarks to the Author):

This is a well written paper focused on investigating the possible effects to dynamic rupture simulation models of an active low-angle normal fault of adding further complications to an already complicated 3D rupture model. The authors investigate the parameter space of adding a more steeply dipping splay of variable dip (both synthetic and antithetic) and the possible effects of shallow off-fault sediment deformation of various strength and thickness in the hanging wall. These results are of some interest to those concerned with how deep fault slip during earthquakes can be potentially manifested at the surface, and especially to those concerned with the degree to which adding elements of increased fault or off-fault complexity can potentially enhance our understanding of how dynamic rupture deformation may be expressed in the shallow geologic record. However, owing to major problems associated with unexplained discrepancies between the previously published reference model (Biemiller et al., 2022) used as the basis for this study, lingering questions regarding model formulation and results, and the tendency for these additional elements of model complication to move the results further away from the limited geologic controls and observations used to infer the first-order characteristics of the rupture model, this tends to severely limit the significance and usefulness of this study, and makes it unpublishable in its current form.

We thank the reviewer for this detailed review, their attention to detail, and the compelling questions and concerns raised about discrepancies between different models. We have done our best to address each comment individually below.

We note that many of the apparent discrepancies between models in Biemiller et al. (2022) and the models in this study stem from differences when defining a suitable reference model. Here, with more realistic modeling ingredients, we design our suite of comparably large earthquake models with slightly different initial conditions compared to the earlier simpler

models. Specifically, models in the present study use more realistic near-hydrostatic pore pressures aligned with those inferred from microstructural observations of the exhumed Mai'iu fault core (Little et al., 2019), which are lower than the moderately overpressured values used in the simpler elastic single-fault models of Biemiller et al. (2022). Lower pore pressure effectively strengthens the main fault and thus requires stronger nucleation over stresses to initiate sustained dynamic ruptures.

Most models (1,2,4; Table 1) in Biemiller et al. (2022) impose a temporary overstress with a maximum magnitude of 30 MPa over a region with a radius of 2 km which produces realistic magnitudes for the simple purely elastic single-fault model with moderately overpressured pore pressure. In the more complex models of the present study, which consider more realistic pore pressure conditions and additional dynamic processes like slip on pre-existing splay faults and/or coseismic off-fault deformation, all modeled ruptures are nucleated with stronger over stresses of up to 45 MPa over a region with a radius of 3 km.

Observational constraints on earthquake initiation processes are elusive and earthquake cycle simulations that incorporate spontaneous (aseismic) nucleation are methodologically and computationally currently infeasible at the same level of complexity as 3D dynamic rupture simulations (e.g., Jiang et al., 2022; Uphoff et al., GJI 2022). The sensitivity of dynamic rupture to rupture nucleation has been extensively studied (e.g., Festa and Vilotte, 2006; Shi et al., 2008; Lu et al., 2009; Bizzarri, 2010; Gabriel et al., 2012; 2013 and others). While we have made every effort to construct the simulations in both studies with observationally constrained parameters representative of the Mai'iu fault, the nucleation conditions remain some of the least observationally constrained components of these models: only the nucleation depth of 10-12 km can be constrained by microstructures in the exhumed fault rock sequence (i.e. Ar/Ar minimum ages of pseudotachylites indicate seismic slip down to at least 10-12 km depth) and paleopiezometric analyses of exhumed footwall rocks (i.e. paleostress and fault strength peak @ 10-12 km depth). The location along-strike as well as the magnitude, duration, and areal extent of the rupture nucleation process are difficult to constrain given that no large earthquakes have occurred on this fault during the modern instrumental record.

We now discuss this lack of strong constraints on nucleation conditions and refer to Biemiller et al. (2022) and the sets of models testing different nucleation conditions and locations (Supp. Info. Figs. S2, S3, S4, S6, S7; Table S1) shown in the Supporting Information. Nucleating rupture in dynamic rupture simulations can be done in a variety of ways, such as temporarily reducing fault friction near the hypocenter (e.g., Harris et al., 2021) or temporarily increasing on-fault shear stresses near the hypocenter (e.g., Galis et al., 2015; Biemiller et al., 2022). Beyond selecting one of these implementation methodologies, various strategies exist for assigning values for parameters of the nucleation process such as magnitude, duration, and spatial extent. Facing few constraints on these parameters for the Mai'iu fault, in Biemiller et al. (2022) we aimed to minimize the nucleation procedure by finding and implementing the smallest overstress magnitudes and nucleation radii that generate self-sustained spontaneous dynamic ruptures in each model configuration, as has been done in previous studies such as Galis et al. (2015). A benefit of this approach is that it minimizes the influence of the potentially ill-constrained nucleation procedure on the subsequent stages of the simulation.

This assumption implies that interseismic elastic shear stresses build up in an orderly manner that promotes 'minimal' ruptures, accumulating in a localized region around the

optimal hypocentral region and never exceeding the minimum stress concentration needed to nucleate the smallest possible rupture the fault is capable of hosting. Of course, natural faults are heterogeneous in material, strength, geometry, slip and stress history, etc., and it is possible (or even common) for heterogeneous faults to accumulate heterogeneous sub-critical-to-critical interseismic stresses and strains that eventually contribute to larger-than-minimal earthquake ruptures (e.g., Fletcher et al., 2016; as illustrated in the creep-rate-derived initial stresses in the Rodgers Creek-Hayward-Calaveras Faults dynamic rupture model setups of Harris et al., 2021). All of this to say that the models in Biemiller et al. (2022) examined minimal large ruptures of the Mai'iu fault within a simplified purely elastic Earth with no other pre-existing faults.

We have added a paragraph discussing the uncertainties and implications of imposed nucleation conditions to the new section of the main text titled 'Model Uncertainty & Comparison with Observations':

“Nonetheless, we acknowledge that compared to dynamic rupture modeling studies of well-documented earthquakes, our models are limited by the absence of seismological data from a modern analogue earthquake, leading to fewer constraints on and higher uncertainties in instrumentally observable rupture characteristics like hypocentral location, stress drop, and rupture velocity. Furthermore, dynamically simulating paleoseismic ruptures involves facing the same uncertainties and modeling challenges inherent to all dynamic rupture models. For example, although the sensitivity of dynamic rupture to nucleation characteristics has been extensively studied (e.g., Festa and Vilotte, 2006; Shi et al., 2008; Lu et al., 2009; Bizzarri, 2010; Gabriel et al., 2012; 2013), observational constraints on earthquake initiation processes remain elusive and the conditions imposed to nucleate ruptures remain some of the less well-constrained components of dynamic rupture models. Nucleation in dynamic rupture simulations can be achieved in a variety of ways, such as by temporarily reducing fault friction near the hypocenter (e.g., Harris et al., 2021) or temporarily increasing on-fault shear stresses near the hypocenter (e.g., Galis et al., 2015; Biemiller et al., 2022). Beyond implementing one of these methodologies, various strategies exist for assigning values for parameters of the nucleation process such as magnitude, duration, and spatial extent. Facing few observational constraints on nucleation apart from the pseudotachylite-derived depth of ~10-12 km (Little et al., 2019), Biemiller et al. (2022) found and implemented the smallest overstress magnitudes and nucleation radii that generated self-sustained spontaneous dynamic ruptures in each model configuration, as done in previous studies such as Galis et al. (2015) and illustrated in the Supporting Information section of Biemiller et al. (2022) (Figures S2, S3, S4, S6, S7; Table S1). A benefit of this approach is that it minimizes the influence of the potentially ill-constrained nucleation process on the subsequent stages of the simulation. However, implicit to this nucleation procedure is an assumption that interseismic elastic shear stresses should build up in an orderly manner that promotes ‘minimal’ ruptures, accumulating in a localized region around the optimal hypocentral region and never exceeding the minimum stress concentration needed to nucleate the smallest possible rupture the fault is capable of hosting. In contrast, natural faults are heterogeneous in material, strength, geometry, stress and slip history: thus, it is possible or even common for faults to accumulate heterogeneous sub-critical-to-critical interseismic stresses and strains that eventually contribute to larger-than-minimal earthquake

ruptures (e.g., Fletcher et al., 2016; or as illustrated in the creep-rate-derived initial stresses in the Rodgers Creek-Hayward-Calaveras Faults dynamic rupture model setups of Harris et al., 2021). While the purely elastic single-fault models of Biemiller et al. (2022) examined the minimal dynamic ruptures that could have generated coral-recorded 1-m-scale episodic coastal displacements, our new models (Figures 2, 3, 4) show how such offsets could reflect larger events that dissipate rupture energy via a broader array of shallow deformation mechanisms including splay fault slip and off-fault damage.”

In contrast, our new models in this study show that realistic complexities of the Earth, such as pre-existing splay faults or the finite plastic yield strength of surrounding crustal rocks, tend to dissipate rupture energy from the main low-angle normal fault (via splay fault slip and/or off-fault damage). Under more realistic conditions than the single-fault overpressured elastic model presented in Biemieller et al., 2022, rupture dynamics scenarios thus require higher nucleation energy and/or stronger background stresses to generate similarly sized sustained spontaneous rupture. Stronger nucleation stresses may be interpreted as larger interseismically accumulated and sustained elastic shear stresses, which could arise due to stronger locking or longer interseismic periods (and thus, longer recurrence intervals). Stronger background or initial shear stresses on the fault could correspond either to closer-to-critical interseismic stress accumulation or stronger fault strength, either due to higher static friction or lower pore fluid pressure, as in the models of the present study. These models use lower, near-hydrostatic values of pore fluid pressure (as reported for the Mai’iu fault by Little et al., 2019) and are thus more like the models in Figure S2 of Biemiller et al. (2022) than to the models in the main text.

Although parameters for the larger, stronger nucleation in these models were given in the original manuscript (lines 493-494), we recognize that their influence on the resulting modeled ruptures should have been highlighted and addressed more clearly. We have now revised the sentences describing the nucleation procedure to read:

“Additionally, the lower pore pressures in our models necessitate larger overstresses to initiate sustained dynamic rupture than those used in the purely elastic single-fault models of Biemiller et al. (2022). Here we impose a stronger nucleation than in Biemiller et al. (2022), consisting of a maximum traction perturbation of $T_{nuc} = 45$ MPa over a spherical region of radius $r_{nuc} = 3$ km centered on the fault at 11 km depth for a duration of $t_{nuc} = 1$ s.”

Furthermore, we add new Supplementary Figures S7 and S8 to more clearly illustrate how the strength of the nucleation and the background pore pressure affect splay-detachment dynamics in these models by comparing the model from Figure 3B of this manuscript to one with the weaker nucleation conditions and higher pore pressures used in most models of Biemiller et al. (2022). We also add Supplementary Figure S9 to illustrate that the smaller nucleation overstresses and radius fail to initiate sustained rupture under the lower near-hydrostatic pore fluid pressure conditions used throughout the present study. Figures S7-S9 are reproduced below:

Right: model with 60°-dipping splay fault from Figure 3B
 Left: identical model except with higher pore pressure and a weaker nucleation

Figure S7. Right column.) Evolution of total slip with 0.5 s rupture contours for the model with a 60°-dipping synthetic splay fault (Figure 3B); Left column.) Evolution of total slip with 0.5 s rupture contours for a model with identical geometry and parameters except with higher pore fluid pressure ($\lambda_f = 0.66$) and weaker nucleation stresses (up to 30 MPa) applied over a smaller hypocentral area of radius 2 km.

Right: model with 60°-dipping splay fault from Figure 3B
 Left: identical model except with higher pore pressure and a weaker nucleation

Figure S8. Right column.) Dip-slip rate for the model with a 60°-dipping synthetic splay fault (Figure 3B); Left column.) Dip-slip rate for a model with identical geometry and parameters except with higher pore fluid pressure ($\lambda f = 0.66$) and weaker nucleation stresses (up to 30 MPa) applied over a smaller hypocentral area of radius 2 km.

A. Low pore pressure ($\lambda_f = 0.44$) & weak nucleation
overstress: $\tau_{\text{nuc}} = 30$ MPa with radius $r_{\text{nuc}} = 2$ km
60°-dipping synthetic splay fault; No plasticity

C. Same as A., but with off-fault plasticity

B. Transient downdip shear stress (MPa)
halfway through nucleation phase:

Zoomed view of hypocentral region:

Figure S9. A.) Peak slip-rate and total slip for an elastic model with the near-hydrostatic pore fluid pressures ($\lambda_f = 0.44$) used throughout this study and the weaker nucleation conditions used in the previous single-fault elastic models of Biemiller et al. (2022), with over stresses up to 30 MPa applied over a smaller hypocentral area of radius 2 km, as shown by the plot of transient downdip shear stress at $t = 0.5$ s in B. These stress conditions fail to nucleate sustained dynamic rupture. C.) Dynamic rupture similarly fails to nucleate in a model identical to that in A and B but with off-fault plasticity enabled.”

Harris, R. A., Barall, M., Lockner, D. A., Moore, D. E., Ponce, D. A., Graymer, R. W., Funning, G., Morrow, C. A., Kyriakopoulos, C., & Eberhart-Phillips, D. (2021). A Geology and Geodesy Based Model of Dynamic Earthquake Rupture on the Rodgers Creek-Hayward-Calaveras Fault System, California. *Journal of Geophysical Research: Solid Earth*, 126(3), 1–28. <https://doi.org/10.1029/2020JB020577>

Galis, M., Pelties, C., Kristek, J., Moczo, P., Ampuero, J. P., & Mai, P. M. (2015). On the initiation of sustained slip-weakening ruptures by localized stresses. *Geophysical Journal International*, 200(2), 890–909. <https://doi.org/10.1093/gji/ggu436>

Little, T. A., Webber, S. M., Mizera, M., Boulton, C., Oesterle, J., Ellis, S., ... Wallace, L. (2019). Evolution of a rapidly slipping, active low-angle normal fault, Suckling-Dayman metamorphic core complex, SE Papua New Guinea. *Bulletin of the Geological Society of America*, 131(7–8). <https://doi.org/10.1130/B35051.1>

Major discrepancies with published reference model as presented in this study

1) Although the color scheme of shaded total slip (m) for the reference model presented in Fig.3g appears identical to the published reference model (Fig.5, Biemiller et al., 2022), there is in fact an unexplained factor of two discrepancy in color bars. Color bars that previously ranged from 0.05 to 5 m now range from 0.05 to 10 m for the same color gradations. This implies that the total slip at depth, that previously ranged from 3-4 m are now 6-7 m for the exact same model even though the resulting magnitude (Mw 7.4) did not change. This is not possible. Although one might suspect that there has been a simple mistake in the color bar assignment for the figure (as it looks like they mistakenly used the color bar for Fig.S1, rather than the correct one from Fig.5), this is apparently not the case. Thus, where previously the total slip at depth was on the order of 3-4 m between 6-13 km depth (Fig.5, Biemiller et al., 2022), the authors now claim that the reference model “ruptures in a Mw 7.4 earthquake with up to 7 m of slip concentrated between 6-13 km depth” (Line 132). The reference model now produces nearly twice as much slip as it did before. This also implies that with the addition of the hanging-wall synthetic splay fault, total slip at the depth increases up to 8-9 m in the same region of the deep detachment. What is the physics of this dramatic doubling of the total slip on the deep detachment?

We again thank the reviewer for catching this apparent discrepancy and refer them to our previous response, including the revised text and new supplementary figures incorporated in the revised manuscript. Models in this study used a wider nucleation region combined with stronger overstress and lower, near-hydrostatic pore fluid pressures; therefore, they are most similar to the model in Figure S2c of Biemiller et al. (2022) which hosts ~ 7 m of slip near the hypocenter (reproduced below).

C.

$$\lambda_f = 0.44; R_0 = 0.8; T_{\text{nuc}} = 45 \text{ MPa}; r_{\text{nuc}} = 3 \text{ km}$$

Figure S2c of Biemiller et al. (2022).

Increased slip on the deeper detachment in the presence of synthetic splay faults occurs during a second phase of slip resulting from back-propagating rupture reflected at the free surface (e.g., Dunham, 2005) downdip from the free surface. We add new plots in Supplementary Figures S7 and S8 (shown below) to illustrate this process and revise the main text to explain the source of this increased slip:

“Rupture to the surface on synthetic splays generates a strong back-propagating free-surface-reflected rupture front (e.g., Dunham, 2005) that drives a secondary phase of slip on the deeper LANF (Figures S7 & S8), resulting in higher total slip and larger earthquakes (Figure 3A-C).”

Dunham, E. M., 2005: Dissipative interface waves and the transient response of a three-dimensional sliding interface with Coulomb friction, *J. Mech. Phys. Sol.*, 53, 327–357, doi:10.1016/j.jmps.2004.07.003

Right: model with 60°-dipping splay fault from Figure 3B
 Left: identical model except with higher pore pressure and a weaker nucleation

“Figure S7. Right column.) Evolution of total slip with 0.5 s rupture contours for the model with a 60°-dipping synthetic splay fault (Figure 3B); Left column.) Evolution of total slip with 0.5 s rupture contours for a model with identical geometry and parameters except with higher pore fluid pressure ($\lambda_f = 0.66$) and weaker nucleation stresses (up to 30 MPa) applied over a smaller hypocentral area of radius 2 km.

Right: model with 60°-dipping splay fault from Figure 3B
 Left: identical model except with higher pore pressure and a weaker nucleation

Figure S8. Right column.) Dip-slip rate for the model with a 60°-dipping synthetic splay fault (Figure 3B); Left column.) Dip-slip rate for a model with identical geometry and parameters except with higher pore fluid pressure ($\lambda_f = 0.66$) and weaker nucleation stresses (up to 30 MPa) applied over a smaller hypocentral area of radius 2 km.”

Given that both the slip and fault rupture area have increased significantly owing the presence of a synthetic splay fault, the moment and magnitude must also increase accordingly from Mw 7.4 to Mw 7.8, but the authors claim it is only Mw 7.5.

We appreciate this concern over potential discrepancies in the correlations between slip, rupture area, and reported moment magnitude between the models with and without splays. This comment prompted us to check and verify our moment and moment magnitude calculations for these models. We note that we calculate seismic moments directly from the modeled fault rupture dynamics. The reference model without splays in Figure 3G hosts a Mw 7.4 (rounded from Mw 7.41174) rupture with a moment of $1.67e20$ Nm. To verify the multi-fault magnitudes, we separated the main fault and splay fault outputs and calculated their moments and magnitudes independently. For example, the reported magnitude for the model with a synthetic 60-degree-dipping splay in Figure 3B is Mw 7.5:

Rebuttal Letter Figure 1. Plot of the distinct surfaces used for separate calculations of the moment contribution from slip on the main fault (blue) and the splay fault (red).

The separate calculations show that the main fault moment of $1.994e20$ Nm (equivalent to a Mw 7.463) and the splay fault moment of $4.205e19$ Nm (equivalent to a Mw 7.013) sum to a total moment of $2.4145e20$ Nm and a total Mw 7.519 for this event. Despite a few meters more slip on the main fault and a few meters of slip on the splay faults, the models with synthetic splays (Figure 3A-C) achieve only modestly larger moments and magnitudes than the reference model, in part due to the reduction of slip on the shallower portions of the main fault above 5 km depth.

2) The published reference 3D model does not explicitly document the change in dip along strike or down-dip. The reader might assume that the 2D geometry shown in Fig.1d (Biemiller et al., 2022) might apply for most of the reference model, but this is apparently not the case. Although it is difficult to tell from the available 3D perspective views, the eastern edge of the model tends to show a smooth curved detachment that steepens from near 20° at the surface to about 30° at depth (Fig.3a, Biemiller et al., 2022). The more steeply dipping section used for Goodenough Bay that allows rupture to propagate to shallow depths apparently dips at 35° and this steeper dip extends to depths of 5 km or more. The problem is that, although it is again difficult to tell from the limited 3D perspective views provided, it appears that for the reference

model used in this study, there is a change of $\sim 30^\circ$ in the dip of the shallow detachment at about 2 km depth for the far eastern end of the model from a nearly planar deeper detachment that dips at about 20° to a shallow more steeply dipping segment that dips at about 50° (Fig. 2E & 2F, this study). This change in model geometry relative to the original reference model (Fig. 3a, Biemiller et al., 2022) is not explained or even identified.

We thank the reviewer for bringing to our attention the misleadingly steep appearance of the oblique along-strike views of the modeled fault in Figures 2E & 2F. The modeled Mai'iu fault geometry used in this study is identical to that used in Biemiller et al. (2022) and includes the same steeper near-surface Goodenough segment that allows limited surface rupture along this portion of the fault in some models, as shown and discussed in Biemiller et al. (2022). The misleading steepness of certain segments in Figures 2E & 2F are most likely an unintentional consequence of the camera angle chosen to highlight all 6 different splay fault orientations in 2E. Showing these splay fault geometries remains important; thus, to address this comment and clarify the fault geometry, we add an additional Supplementary Figure S10 with different views of the modeled fault geometry (reproduced below) and add these lines to the end of the caption for Figure 2:

“Note that although the oblique views in E & F may cause portions of the Mai'iu fault to appear abnormally steep, the modeled fault geometry is identical to that described and shown in Biemiller et al. (2022) (Section 2.3.1; Fig. 3), derived from that of Webber et al. (2020). Supplementary Figure S10 further illustrates the adapted fault geometries used in our simulations.”

Looking due West:

Looking Northwest along-strike of the splay:

Looking Southwest from above:

Mapview:

“Figure S10. Additional plots of the model with a 60°-dipping synthetic splay fault (Figure 3B) illustrating the prescribed fault geometries. Left: different view angles with main and splay fault labeled. Right: modeled faults viewed from above but lit with different lighting options.”

Lingering questions of model formulation and results

3) From the original formulation of the reference model, total slip at depth is about 3-4 m on a fault that dips at 30°. This produces a maximum vertical displacement (subsidence) of 2 m. However, if this displacement is buried at depths of 6-13 km, it is unlikely to be fully expressed at the surface. Buried or blind ruptures with meters of displacement (e.g., Northridge, Loma Prieta, etc.) rarely produce surface displacements more than a few 10's of centimeters. However, the authors assert that such buried fault slip will yield vertical surface displacements on the order of 1 m or more (Fig.5, Biemiller et al., 2022). Although I am not an expert in modeling geodetic changes resulting from buried fault slip, this still does not seem realistic to me. Yes, if this slip reaches the surface, as it is modeled to do in Goodenough Bay, one might expect up to 1.1 m of related vertical surface displacement, but not farther outboard above where the fault slip is buried at depths of 6 km or more. The authors thus tend to imply that whether fault slip reaches the surface or is buried at depths below 5 km, the vertical surface displacement is the same. This does not make sense to me.

We thank the reviewer for raising this issue and agree that similar increments of buried vs. surface-rupturing slip should not cause the same magnitude of vertical surface displacement. We believe that this issue of the apparent along-strike similarity of the resulting subsidence

increments in the reference model of Biemiller et al. (2022) arose from the regrettably vague colorbar and partial transparency used to plot vertical surface displacements in Figure 5b of that paper. In that model, vertical surface displacements across the fault only approach 1.2 m along the Goodenough segment where slip reaches the surface. Further along-strike and outboard (NW), the subsided area above the buried slip patch is actually (subtly) lighter blue, corresponding to lower subsidence of < 0.8 m as noted in the annotated clip from Figure 5b below, along with a new plot of vertical surface displacement on a clearer color scale. Buried slip alone produces significantly smaller vertical surface displacements than slip along surface-rupturing segments, as expected. We reproduce Figure 5b from Biemiller et al. (2022) directly (below, left) and replot the exact same model from that figure on a clearer color scale (below, right) to show more clearly how the resulting surface displacements varied along-strike:

Rebuttal Letter Figure 2. Left: subsidence plot from Figure 5b of Biemiller et al. (2022) annotated to highlight decreased subsidence near the buried rupture portion of the fault and greater subsidence near the surface-rupturing portion. Right: vertical displacement (m) for the exact same model plotted on a different color scale to highlight these same features.

Apparent discrepancies between the new models and those of Biemiller et al. (2022) should be clarified by the additional descriptions we have added throughout the revised manuscript distinguishing the two unique sets of models and their parameterizations. In particular, the new section and paragraph on nucleation conditions should clear up any ambiguity between these groups of models:

“While the purely elastic single-fault models of Biemiller et al. (2022) examined the minimal ruptures that could have generated coral-recorded 1-m-scale episodic coastal displacements, our new models (Figures 2, 3, 4) illustrate how such offsets could reflect larger events that dissipate rupture energy via a broader array of shallow deformation mechanisms including splay fault slip and off-fault damage.”

4) For the added complication of a hanging-wall splay fault in this study, although it is certainly interesting to see the results of investigating the parameter space of variable dip and dip

direction (from synthetic to antithetic), I was curious as to why the authors did not use the locations and dips of the actual hanging-wall splay faults imaged, mapped and defined at depth with MCS reflection data (e.g., Lines 1190, 1191, 1193, Fitz and Mann, 2013 a,b). These results suggest few if any mapped splays occupy the seafloor outcrop positions proposed for the modeled splay fault, or would even project to intersect the detachment at 5 km depth, as all the proposed modeled splay faults in this study do. There is also the issue that most of the mapped splay faults located farther west on the western side of the detachment (e.g., Line 1181) are inactive and are not imaged to offset Pleistocene and younger sediments.

The question of whether to model specific splay faults imaged above the Mai'iu fault or a broader set of representative and systematically varying splay faults is an important one that we considered while designing our experiments. We decided for the more generic splay fault representations, which we further explain and justify later in this response. First, we review evidence for or against the presence of splay faults above the Mai'iu fault with similar geometries to those in our models. To us it is not clear that the seismic reflection data referenced in this comment (Fitz & Mann, 2013a, 2013b) are strictly inconsistent with ~5-km-deep splay faults above the Mai'iu fault. In the eastern portion of the fault system in the Goodenough Basin, those authors interpret variably dipping steeper normal faults in the hanging wall of the Owen-Stanley / Mai'iu fault penetrating to depths >5 km:

Rebuttal Letter Figure 3. Figures from Fitz & Mann (2013a, 2013b) illustrating the seismically imaged and interpreted crustal structures in the Goodenough Basin above the Owen-Stanley/Mai'iu fault. Top: map showing locations of these lines. Panels 2 & 3: schematic structural interpretation showing normal faults in the immediate hanging wall of the shallowly dipping Owen-Stanley fault zone penetrating to >5 km depth. Panels 4 & 5: interpreted seismic lines showing similarly deep-seated conjugate normal faults in the Goshen fault zone.

Considering these surveys did not extend close enough to the shoreline to image the Mai'iu fault itself, it is difficult to determine with much confidence whether any of such hanging wall faults approach the Mai'iu fault around 5 km depth, though it seems possible they do. In fact, in the western Goodenough Basin near Cape Vogel where the surveys reached closer to the shoreline, Fitz (2011) interprets some of the most nearshore steeply dipping normal faults as active faults soling into an inferred shallowly dipping structure at depth (4-5 s TWT):

Rebuttal Letter Figure 4. Structural interpretation from Fitz (2011) showing similarly deep-seated normal faults soling into the inferred underlying shallow detachment fault from 3-5 s TWT.

West of Line 1168, the survey lines certainly do not extend close enough to shore to image the presence or absence of splays above the Mai'iu fault. In fact, the most recent structural models of the Mai'iu fault surface place its 5 km depth contour onshore or just offshore, directly below the coastline North of Mt. Dayman, Mt. Gwoira, and Cape Vogel (Left: Webber et al., 2020; Right: Mizera et al., 2021):

Rebuttal Letter Figure 5. Recently published structural models of the Mai'iu fault geometry (Left: Webber et al., 2020; Right: Mizera et al., 2021) showing the onshore or nearshore position of the 5 km depth contour.

Antithetic splay faults intersecting the Mai'iu fault at 5 km depth would breach the surface just offshore, well beyond the Southward extent of these reflection images, while 5-km-deep synthetic splay faults would outcrop onshore in the hanging wall lowlands of Cape Vogel and the coastal plains North of Mt. Dayman, where many active normal fault scarps have been mapped (Little et al., 2019):

Rebuttal Letter Figure 6. Abundant active normal faults in the hanging wall of the Mai'iu fault mapped by Little et al. (2019).

Further West, mapped steeply dipping normal faults penetrating to similar depths appear to have aided shallow LANF abandonment and dissected the footwall and the inactive abandoned shallow portion of this segment of the Mai'iu fault.

Nonetheless, rather than constructing models with splay fault geometries that mirror specific seismically imaged splays of the Mai'iu fault, we instead explore the range of possible splay fault geometries above low-angle normal faults to better understand the dynamic interactions between rupture of variably-dipping splays and the underlying detachment fault. To standardize the implementation of splays with different dips, we chose a uniform depth at which the splays each intersect the detachment. Setting that intersection depth to 5 km ensures that our models capture the dynamic behavior of some of the largest, most prominent and demonstrably active splay faults from well-documented active LANF systems like the Altotiberina fault (e.g., Valoroso et al., 2017; Marzorati et al., 2014; Vuan et al., 2020; Essing & Poli, 2022). For example, the interplay between clustered seismicity and aseismic slip on the shallowly dipping Altotiberina fault and the steeply dipping Gubbio splay fault in its hanging wall is one of the best-documented cases of slip-induced stress transfer between a LANF and its

splay fault (e.g., Valoroso et al., 2017; Vuan et al., 2020; Essing & Poli, 2022). The partially creeping and microseismically active antithetic Gubbio fault (e.g., Anderlini et al., 2016; Valoroso et al., 2017) joins the Altotiberina fault at ~5 km depth and, given the Gubbio fault's well-documented activity and sensitivity to Altotiberina fault slip, it is amongst the LANF-bounded-splays most likely to slip coseismically during a large LANF rupture. Furthermore, another steeply dipping microseismically active splay fault synthetic to the Altotiberina intersects it at similar depths of 5-7 km (e.g., Marzorati et al., 2014, Figure 7), obviating our interest in modeling rupture of both synthetic and antithetic splays that intersect the LANF at similar depths.

To summarize, rather than attempting to recreate the exact geometries of seismically imaged faults above the Mai'iu fault, these models include splay faults penetrating to 5 km because these geometries seem most representative of the major demonstrably active splay faults above active LANFs worldwide. Nonetheless, geologic and geophysical observations of the Mai'iu fault do not appear inconsistent with the presence of active hanging wall faults penetrating to ~5 km depth and possibly intersecting the Mai'iu fault.

In the revised manuscript, we have added considerations of these issues and an outlook for future work with more complex and specific splay fault geometries to the new discussion section:

“Finally, we note that the systematically dip-variable planar splay faults in our models are broadly representative of the most demonstrably seismically active splay faults in LANF systems like those above the Altotiberina fault (e.g., Marzorati et al., 2014; Anderlini et al., 2016; Valoroso et al., 2017; Vuan et al., 2020; Essing & Poli, 2022), but that future work incorporating the geometries of mapped and seismically imaged splays of the Mai'iu fault (e.g., Fitz & Mann, 2013a, 2013b; Little et al., 2019; Mizera et al., 2019; Webber et al., 2020) into dynamic rupture simulations could generate more targeted models showing how specific splays might be expected to slip in various rupture scenarios of the Mai'iu fault.”

Anderlini, L., Serpelloni, E., & Belardinelli, M. E. (2016). Creep and locking of a low-angle normal fault: Insights from the Altotiberina fault in the Northern Apennines (Italy). *Geophysical Research Letters*, 43(9), 4321–4329. <https://doi.org/10.1002/2016GL068604>

Essing, D., & Poli, P. (2022). Spatiotemporal Evolution of the Seismicity in the Alto Tiberina Fault System Revealed by a High-Resolution Template Matching Catalog. *Journal of Geophysical Research: Solid Earth*, 127(10). <https://doi.org/10.1029/2022JB024845>

Fitz, G. (2011). Offshore mapping and modeling of Miocene-Recent extensional basins adjacent to metamorphic gneiss domes of the D'Entrecasteaux Islands, eastern Papua New Guinea. University of Texas at Austin.

Fitz, G., & Mann, P. (2013). Evaluating upper versus lower crustal extension through structural reconstructions and subsidence analysis of basins adjacent to the D'Entrecasteaux Islands, eastern Papua New Guinea. *Geochemistry, Geophysics, Geosystems*, 14(6), 1800–1818. <https://doi.org/10.1002/ggge.20123>

Fitz, G., & Mann, P. (2013). Tectonic uplift mechanism of the Goodenough and Fergusson Island gneiss domes, eastern Papua New Guinea: Constraints from seismic reflection and well data. *Geochemistry, Geophysics, Geosystems*, 14(10), 3969–3995. [https://doi.org/10.1002/GGGE.20208@10.1002/\(ISSN\)1525-2027.LITHOS1](https://doi.org/10.1002/GGGE.20208@10.1002/(ISSN)1525-2027.LITHOS1)

Harris, R. A., Barall, M., Lockner, D. A., Moore, D. E., Ponce, D. A., Graymer, R. W., ... Eberhart-Phillips, D. (2021). A Geology and Geodesy Based Model of Dynamic Earthquake Rupture on the Rodgers Creek-Hayward-Calaveras Fault System, California. *Journal of Geophysical Research: Solid Earth*, 126(3), 1–28. <https://doi.org/10.1029/2020JB020577>

Little, T. A., Webber, S. M., Mizera, M., Boulton, C., Oesterle, J., Ellis, S., ... Wallace, L. (2019). Evolution of a rapidly slipping, active low-angle normal fault, Suckling-Dayman metamorphic core complex, SE Papua New Guinea. *Bulletin of the Geological Society of America*, 131(7–8). <https://doi.org/10.1130/B35051.1>

Marzorati, S., Massa, M., Cattaneo, M., Monachesi, G., & Frapiccini, M. (2014). Very detailed seismic pattern and migration inferred from the April 2010 Pietralunga (northern Italian Apennines) micro-earthquake sequence. *Tectonophysics*, 610, 91–109. <https://doi.org/10.1016/j.tecto.2013.10.014>

Mizera, M., Little, T., Boulton, C., Katzir, Y., Thiagarajan, N., Prior, D. J., ... Smith, E. G. C. (2021). Using Syntectonic Calcite Veins to Reconstruct the Strength Evolution of an Active Low-Angle Normal Fault, Woodlark Rift, SE Papua New Guinea. *Journal of Geophysical Research: Solid Earth*, 126(8). <https://doi.org/10.1029/2021JB021916>

Ramos, M. D., & Huang, Y. (2019). How the Transition Region Along the Cascadia Megathrust Influences Coseismic Behavior: Insights From 2-D Dynamic Rupture Simulations. *Geophysical Research Letters*, 46(4), 1973–1983. <https://doi.org/10.1029/2018GL080812>

Ramos, M. D., Huang, Y., Ulrich, T., Li, D., Gabriel, A. A., & Thomas, A. M. (2021). Assessing Margin-Wide Rupture Behaviors Along the Cascadia Megathrust With 3-D Dynamic Rupture Simulations. *Journal of Geophysical Research: Solid Earth*, 126(7), 1–22. <https://doi.org/10.1029/2021JB022005>

Valoroso, L., Chiaraluce, L., Di Stefano, R., & Monachesi, G. (2017). Mixed-Mode Slip Behavior of the Altotiberina Low-Angle Normal Fault System (Northern Apennines, Italy) through High-Resolution Earthquake Locations and Repeating Events. *Journal of Geophysical Research: Solid Earth*, 122(12), 10,220–10,240. <https://doi.org/10.1002/2017JB014607>

Vuan, A., Brondi, P., Sukan, M., Chiaraluce, L., Di Stefano, R., & Michele, M. (2020). Intermittent Slip Along the Alto Tiberina Low-Angle Normal Fault in Central Italy. *Geophysical Research Letters*, 47(17), 1–11. <https://doi.org/10.1029/2020GL089039>

Webber, S., Little, T. A., Norton, K. P., Österle, J., Mizera, M., Seward, D., & Holden, G. (2020). Progressive back-warping of a rider block atop an actively exhuming, continental low-angle normal fault. *Journal of Structural Geology*, 130(May 2019). <https://doi.org/10.1016/j.jsg.2019.103906>

5) In terms of the dynamic behavior of the more steeply dipping splay faults, the question is whether these faults with little cumulative offset would be similar in terms of static friction, strength and rupture behavior to the detachment with known large cumulative offset, low static friction, and velocity-strengthening materials for shallow fault rocks. It seems that the authors assume the splay has the same characteristics as the detachment at shallow (<5 km) depths. This does not seem likely to me given the steeply dipping splays have significantly less cumulative offset, are younger, and often cut different lithology than the detachment. Thus, although the more steeply dipping splay may be more favorably oriented for normal slip relative to the regional stress field, its other characteristics may be such that slip on the weak low-angle detachment may still be favored to some extent.

We appreciate this insightful comment which raises the important and thought-provoking topic of how fault maturity influences earthquake rupture, particularly in structurally complex systems with multiple generations of faults. In response we clarify the effective frictional strength in our fast-velocity-weakening rate-and-state framework, argue that similar frictional parameterizations of the main and splay faults are a reasonable approximation given how vastly different the interseismic loading conditions would be if fault strength were modulated by the weak, stable gouges in the shallow fault zone, and finally construct and analyze 5 new models with different frictional and stress conditions to test these effects. The new models (included in the revised version as Supplementary Figure S11) illustrate that although shallow LANF friction has some influence on shallow slip behavior, that influence is minor relative to the dominant effects of fault geometry, as concluded in the original manuscript.

We agree that a long-lived detachment fault hosts different lithologies, fault zone (micro)structures, and frictional properties than younger splay faults with limited offset. Both brittle and ductile weakening, resulting from protracted slip and creep over many earthquake cycles, are key processes facilitating the intense localization that allows these low-angle normal faults to persist over millions of years. Recognizing this synextensional fault weakening as a major mechanical feature of such faults, we explored how reduced frictional strength in the upper brittle fault zone influences low-angle normal fault rupture in model 4 of Biemiller et al. (2022). In that model, the frictional strength of the fault was reduced by linearly decreasing the effective static friction coefficient (f_0) from 0.6 to 0.25 from 12 km depth to the surface and accordingly adjusting the magnitude (but not direction) of the applied Andersonian stresses. Interestingly, these heterogeneous interseismic stress accumulation conditions prevent rupture from reaching the surface despite the dynamic frictional weakness of the shallow fault, suggesting that coseismic fault strength may be less important than the effective interseismic fault strength that modulates the magnitude of interseismic stresses that can be accumulated without initiating a rupture.

As keenly noted by the reviewer, the detachment and splay faults in the present study are modeled with the same standard reference friction coefficient of 0.6 (approximating an equivalent effective static frictional strength in our rate-state framework). The reviewer's main concern with applying the same mechanical conditions to the detachment and splay faults is that slip on the frictionally weaker and more velocity-strengthening detachment may be mechanically favorable despite the preferable orientation of the steeper splays. The shallow velocity-strengthening gouges in low-angle normal faults should actually inhibit rupture propagation towards the surface on the main detachment, further favoring splay fault rupture. The lower frictional strength of these velocity-strengthening materials should modulate interseismic fault strength by allowing interseismic creep at relatively low stresses. However, the Andersonian stresses applied in the models in this study are computed based on fault strength governed by a homogeneous static friction coefficient of 0.6, which implies that fault locking in the stronger, deeper portions of the fault dominates interseismically, allowing larger interseismic stresses to accumulate even at shallow depths. Given these loading conditions, we believe it is not unreasonable to represent the effective static friction of the entire fault with the friction coefficient governing its interseismic strength. Weakening to the uniformly low dynamic friction of $f_w = 0.2$ at slip rates above $V_w = 0.1$ m/s ensures that all modeled faults are coseismically weak.

To verify these points and directly address the reviewer's comment, we performed 5 new dynamic rupture simulations with variable friction and initial stress conditions based on different levels of fault maturity, which confirm that the influence of shallow LANF frictional strength (or weakness) is minor relative to that of fault geometry. We include plots from these new models in a new Figure S11, with a caption describing and discussing the new results:

- A.** Original model; friction parameters as plotted; initial stress based on $\mu_s = f_0 = 0.6$
- B.** Same as A., but all faults velocity-weakening above 5 km depth; $a = 0.01 \rightarrow a - b = -0.004$

- C.** Main fault: $f_0 = 0.6$ below 15 km depth, linearly decreasing to 0.3 at surface; Splay: $f_0 = 0.6$
- D.** Velocity-weakening splay: same as C., but $a = 0.01 \rightarrow a - b = -0.004$ on the splay fault

- E.** Same as D., but weaker initial stresses calculated from $f_0 = 0.45$
- F.** Main fault: $f_0 = 0.3$ above 12.5 km depth, 0.6 below 15 km; weaker initial stresses calculated from $f_0 = 0.3$

Figure S11. Total slip after 15 s for A.) the model with a 60°-dipping synthetic splay fault (Figure 3B) and B-F.) additional variants of that model isolating the effects of variable fault friction and stress. B.) Model with shallow velocity-weakening friction on both faults results in more total slip and allows rupture to penetrate slightly further updip on the main fault, above the fault intersection at 5 km depth, highlighting the stabilizing effects of clay-rich velocity-strengthening gouges in these fault systems. C-E.) Models with friction and stress conditions designed to test

the mechanical influence of variable levels of fault maturity between the long-lived, high-offset Mai'iu fault and the younger, immature low-offset splay. C.) A weaker main fault with lower effective static friction decreasing above 15 km depth to $f_0 = 0.3$ at the surface results in similar but muted additional slip than in B, suggesting that the strength of the main fault may be less important than its frictional stability in modulating shallow rupture patterns. D.) Model with identical conditions to C except velocity-weakening materials in the splay fault generates indistinguishable slip patterns from those of C, suggesting splay fault slip is largely insensitive to splay fault frictional stability. E.) Model with lower initial stresses calculated based on a weaker coefficient of static friction of 0.45 results in similar patterns of main and splay fault slip but with significantly less total slip, while F.) one with even weaker stresses and static friction of 0.3 fails to nucleate sustained dynamic rupture. E & F suggest that if fault strength were governed by the weak velocity-strengthening gouges in the mature LANF core, interseismic loading would be relieved by creep and thus interseismic stresses could not accumulate to levels capable of producing the large paleoearthquakes recorded in these LANF systems.”

Finally, we note that splay fault rupture in our models appears to occur in direct response to dynamic stress fluctuations unrelated to shallow fault friction, as illustrated in the new Supplementary Figures S6-S8. Rupture jumps ahead of the updip-propagating rupture front to initiate slip on the splay before the main rupture front encounters the splay-detachment intersection or interacts with the potentially weaker shallow portion of the detachment. This dynamic triggering mechanism is comparable to the 2D main-splay fault activation in dynamic rupture models informed from long-term geodynamic subduction simulations recently described in van Zelst et al., 2022. Thus, it does not appear that variations of friction in that part of the LANF would alter the initiation of rupture on the splay fault. This process is further detailed in these lines added to the ‘Discussion & Conclusions’ section of the revised manuscript to address this issue and highlight the new model results of Figure S11:

“We note that although all faults are dynamically weak at high slip rates in these models, we do not account for static frictional weakness of mature gouges in the LANF (see model 4 of Biemiller et al., 2022), which could promote slip on the LANF over younger splays with stronger immature gouges. This strength contrast may enhance LANF slip and reduce splay fault slip relative to the modeled slip distributions in Figure 2. However, features of our modeled ruptures suggest that splay fault slip initiates in response to structurally modulated interactions between dynamic stresses and preferentially oriented pre-existing structures with little regard for shallow fault friction. For example, in the model with a 60°-dipping synthetic splay fault (Figure 2B), dynamic stresses ahead of the deeper rupture clamp the LANF and unclamp the splay, dynamically initiating spontaneous rupture of the splay fault before the main rupture front reached the LANF-splay intersection (Figure S6-S8). Thus, the effects such static frictional strength variations would have on shallow slip partitioning appear minor relative to first-order structural controls on shallow dynamic slip viability such as those exerted by splay fault geometry. Additional models analyzing variable friction and stress parameterizations based on different degrees of fault maturity (Figure S11) support this conclusion.”

6) It is helpful that the first-order model results of adding a hanging-wall splay seem to be consistent with expectations: a) antithetic splays are not particularly favored to localize shallow slip or to alter the slip on detachment, especially for steeper dips; b) on the other hand, synthetic splays tend to localize slip at shallow depths and increase slip on the deeper detachment, likely owing to an increase in dynamic shear stress; c) but increasing dip of the synthetic splay tends to reduce the slip on the deep detachment by increasing the normal (clamping) stress. This is all well and good. Unfortunately, it highlights that adding this additional fault complexity of a hanging-wall splay also tends to move the model results farther away from the limited observations used to define first-order rupture model parameters. Increased model complexity tends to yield results less consistent with observations

We acknowledge that adding model complexity can lead to results that don't match observations of a particular event as well as a simpler model could; however, matching observations is not the main goal of our current study. Instead, we aim to understand how mechanical heterogeneity, pre-existing faults, and inelastic bulk rheologies influence a propagating rupture along a low-angle normal fault. We believe our models accomplish these goals even if they do not match the sparse paleoseismic record of surface subsidence quite as well as simpler elastic models without splay faults.

While matching observations from the coral paleogeodetic record is a valuable target and was explored in our earlier study, it is certainly not the only goal or valuable result of dynamic rupture models. There are enough unconstrained and ill-constrained parameters in earthquake physics and earthquake observations that we may not expect to achieve a unique model that perfectly represents a prehistoric rupture preserved by sparse paleoseismic evidence or the generally heterogeneous physical properties and complex coseismic deformation processes around a given fault for as of yet unobserved earthquakes that may someday occur. While progress has been made for well-instrumented, well-documented faults with a robust history of frequent earthquakes, earthquake modeling is challenging but nevertheless important on a seismogenic fault with no modern large earthquakes located in a remote, minimally instrumented and arguably understudied part of the world.

Our goal with the simpler models of Biemiller et al. (2022) was to see what could be learned about the dynamics of the infrequent and enigmatic ruptures of low-angle normal faults by constructing dynamic rupture simulations with parameters constrained by a wealth of multidisciplinary observations and data. While the coral-recorded subsidence estimates provided a broad target against which to cross-check our simulation results to ensure they were in the right ballpark, matching these emergence estimates was not the main priority. Instead, we analyzed how different principal stress orientations proposed by theory and inferred from analyses of exhumed fault rocks affected rupture characteristics of the resulting events. There simply aren't enough paleoseismic data for it to be worthwhile to match every estimate of paleoseismic uplift and subsidence. Nonetheless, those models showed how data-constrained dynamic rupture simulations might be used to help answer longer-timescale geologic, tectonic, or structural questions in certain settings.

To better explain the goals, uncertainties, and challenges of modeling paleoearthquakes, as well as to further distinguish between this study and previous models, we have added a new discussion section to the revised manuscript titled 'Model Uncertainty & Comparison with Observations,' the second and third paragraphs of which address these

concerns and further explains the goals, challenges, and uncertainties involved in modeling paleoearthquakes. We do not reproduce these new lines here as they have already been reproduced in a previous response.

7) The primary geologic control used to define the reference model parameters are the locations and heights of exhumed, offset footwall corals near the coast of Goodenough Bay (Biemiller et al., 2020, 2022). The inferred average vertical surface displacements (~1.2 m) were used to define the length, width, moment and M_w of the simulated rupture. However, if linearly reduced static friction levels extending from 0.6 at 12 km depth to 0.25 at the surface are imposed on the reference model (Model 4, Biemiller et al., 2022), which seem to be more consistent with the observed fault rocks, the simulated earthquake is reduced to M_w 7.0 and rupture does not penetrate to the surface along any segment. Thus, adding somewhat more realistic static friction levels up-dip along the detachment, tends to decrease the ability of the rupture model to match the observations of the offset corals.

This issue that adding model complexity tends to move the results farther away from the limited geologic controls and observations used to define the model is also a problem in this study that investigates adding the increased complexity of a hanging-wall splay or off-fault sediment deformation. As the authors state: “Modeled coseismic subsidence is larger, more localized, and farther outboard when the hanging wall is cut by synthetic splay faults (Fig. 3A-C) and overlain by thick sedimentary sections (Fig. 4D-F)”. Although the subsidence is larger, the location of surface deformation is much farther offshore where it is not observed, if the exposed, exhumed, coastal offset corals near the outcrop of the detachment are an indication of the principal surface displacement location.

We thank the reviewer for these comments and refer to our previous response outlining the scientific goals of these dynamic rupture models, which can be achieved without perfectly fitting paleoseismic surface displacement data from a particular prehistoric earthquake.

Adding realistic shallow friction helped us learn about the mechanical influence of frictionally weak velocity-strengthening gouges in the shallow fault zone on a propagating rupture. Incorporating realistic friction moves that particular model further away from matching one subsidence datapoint, but this doesn't mean that model fails or is any worse than the original. Although the original reference model matches the Goodenough emergence increments quite well, it is certainly not a unique solution, and it is quite possible that it uses pore fluid pressure or initial stress values that are far from those in the real Mai'iu fault.

We again appreciate the reviewer's comments and perspective, and we acknowledge that constructing and tuning a dynamic rupture simulation to match coseismic observations of a particular earthquake is a common and worthwhile practice for certain fault zones. However, in the case of faults with little-to-no modern seismic activity like the Mai'iu fault or the Cascadia megathrust (e.g., Ramos et al., 2021), we can learn as much from models that don't perfectly fit the paleoseismic data as we can from those that do.

To address these issues in the revised manuscript, the new discussion section 'Model Uncertainty & Comparison With Observations' now more clearly describes the scientific goals and challenges/uncertainties of modeling paleoearthquakes, as well as more clearly distinguishes between the previous models of Biemiller et al. (2022) and the more realistic and

complex models presented in the present study. We do not reproduce the relevant paragraphs here as they have already been reproduced in a previous response.

Ramos, M. D., Huang, Y., Ulrich, T., Li, D., Gabriel, A. A., & Thomas, A. M. (2021). Assessing Margin-Wide Rupture Behaviors Along the Cascadia Megathrust With 3-D Dynamic Rupture Simulations. *Journal of Geophysical Research: Solid Earth*, 126(7), 1–22.

Summary

The major discrepancies in total slip and fault geometry between the published reference model and the version of the same reference model used in this study, the outstanding questions of model formulation and results, the inconsistency between model splay fault geometry and imaged splay faults defined by MCS data, and the tendency for added model complexity of splay faults and/or off-fault deformation to move model results farther away from the limited geologic controls used to define first-order model parameters, severely limit the significance and usefulness of this study, and makes it unpublishable in its current form.

We thank the reviewer for their careful and critical review. We appreciate the comments that identified portions of the original manuscript that were unclear or inadequately explained, which we have clarified in the revised manuscript. We hope that the revised manuscript and our extensive responses help clear up the perceived model discrepancies and explain how these models match the scope of the scientific goals of this study.

Reviewer #2 (Remarks to the Author):

Review of “Duelling dynamics: competition between detachment rupture, splay faults, & off-fault damage”

By Biemiller et al.

Submitted to Nature Communications

Review by Andrew Meigs

Key Results

This study addresses the impact that (1) fault splays and (2) off-fault damage have on rupture of a low angle normal fault (LANF). Dynamic rupture simulations explore the resultant slip using a model based on an extensively studied, archetypal low angle normal fault in New Guinea.

Synthetic fault splays (synthetic = dip in the same direction as the master LANF; antithetic = dip in the opposite direction of the LANF) tend to enable connectivity between the master fault and splays and to promote slip to the surface in the simulations. Antithetic splays impede connectivity and rupture to the surface. Off fault damage is explored by comparing models with a strong footwall and thin and thick hanging wall basin fill of variable sediment strength.

Distributed damage is associated with weak and intermediate strength sediment and increase in seismic moment relative to a reference model. Models with thick sediment yield coseismic hanging wall subsidence and up to 18% greater seismic moment than either the reference model or models with strong sediment.

Validity

Whereas I am not a modeler, the method appears robust as it employs published code and inputs, the 2 families of models explore topical and appropriate structural and rheological scenarios and the results are conservatively interpreted. I especially appreciated supplemental figure S2-4, which compared the effect of color scheme on the appearance of model runs.

Significance

This study is significant because it builds on an outstanding body of observations, lab measurements and other data from the Mai'iu fault in New Guinea. The fault is one of the most comprehensively studied low angle normal faults in terms of its structures on all scales, its structural evolution, the rheological characteristics of the exhumed fault zone rocks and numerous other aspects of its geological development. Thus, prior studies provide key constraints on the mechanical and rheological boundary conditions of the modeling. My understanding is that the dynamic rupture modeling utilized in this analysis is state of the art. Because so few low angle normal fault earthquakes have demonstrably ruptured seismically, this study provides synthetic outcomes that forecast potential rupture and deformation scenarios in a future earthquake. That future event will provide the sort of data to both calibrate the model and assess its applicability. The three most striking results from my point of view are (1) that synthetic splay faulting is a potential prerequisite to surface rupture, (2) that antithetic faults potential represent barriers to slip transfer and earthquake size and (3) that low basin sediment strength likely plays a role in seismic moment, in positive feedback for basin subsidence and new splay fault formation due low strength sediment's potential for enhanced off-fault damage.

Data and methodology

There aren't really any data presented in the study. The one piece of data is a small photo of a coral outcrop and a relative sea level curve derived from age data from that outcrop.

Analytical approach

I am not qualified to assess the analytical approach in detail. The conceptual framework, variables and concepts are consistent with my understanding of dynamic and static friction, low angle normal fault geometrical and rheological considerations, and coseismic slip distributions on faults.

We thank the reviewer for their time and effort on this thoughtful and constructive review. The reviewer noted some concerns with the original text and provided some suggested improvements, which we address individually below.

Suggested improvements

1) Restrict your terminology to "low angle normal fault" and do not use the "detachment" synonym. Whereas both terms appear in the literature, low angle normal faulting is what you seek to understand and is a clearer and more specific description.

We appreciate this suggestion and agree that 'low-angle normal fault' is the most accurate term here. 'Detachment' was used to invoke a connection between splays in low angle normal fault systems and those above other major crustal-scale plate-boundary faults like subduction megathrusts, as the reviewer noticed in their following comment. We have now replaced 'detachment' with 'low-angle normal fault' or 'LANF' throughout the text and in the title.

2) Clarify why you include so much discussion of subduction megathrust models and earthquakes. Are they included because splay faulting is topical for those kinds of events, because dynamic rupture modeling has focused on that setting or for some other reason? In the case of megathrust events, I understand that the role of splay faults in tsunamigenesis is a hot topic and that a low angle normal fault in the marine realm has potential to generate a tsunami. That said, I am unsure of the importance of that outcome, particularly considering that the Mai'iu fault and splays crop out onshore according to the maps in Little et al. (2019).

We thank the reviewer for alerting us that the paragraph discussing coseismic splay fault behavior in subduction zones felt disjointed within a study focused on normal faults. Subduction zones have the unenviable advantage of hosting more frequent, larger earthquakes than low-angle normal faults. Denser instrumentation paired with more frequent large earthquakes means that the coseismic activity of splay faults in subduction zones are better-documented, more heavily studied, and more extensively modeled with techniques such as dynamic rupture modeling. Thus, most of the recent observations and advances in understanding the mechanics of shallow coseismic splay fault reactivation have come from subduction zones. Thus, we included the paragraph about subduction zone splays to highlight that coseismic splay fault slip and significant shallow off-fault deformation have been observed and modeled in some settings, although examples from low-angle normal faults are less common. To clarify why this paragraph is included in the text, we have revised the second line of this paragraph so that it now reads:

“Coseismic splay fault activity has been documented and modeled most extensively in subduction zones, where splay fault slip and inelastic deformation in the frontal wedge are intensely debated mechanisms that may amplify coseismic seafloor displacements and resulting tsunami heights of megathrust earthquakes like Tohoku-Oki (Ito et al., 2011; Ide et al., 2011; Tsuji et al., 2011; Ma & Nie, 2019; van Zelst et al., 2022).”

3) Be more holistic in your reporting of active low angle normal faults and earthquake events on them. The statement on lines 61-62 that focuses on the absence of LANF events > Mw 7, fails to highlight that a number of M6 events are documented in papers such as Axen (1999), Abers et al. (1997), Abbott et al (2001), Bernard et al. (1997) and Wernicke (1995).

Abers, G. A., Mutter, C. Z., and Fang, J., 1997, Shallow dips of normal faults during rapid extension: Earthquakes in the Woodlark-D'Entrecasteaux rift system, Papua New Guinea: *Journal of Geophysical Research: Solid Earth*, v. 102, no. B7, p. 15301-15317.

Abbott, R. E., Louie, J. N., Caskey, S. J., and Pullammanappallil, S., 2001, Geophysical confirmation of low-angle normal slip on the historically active Dixie Valley fault, Nevada: *Journal of Geophysical Research*, v. 106, no. B3, p. 4169-4981.

Axen, G. J., 1999, Low-angle normal fault earthquakes and triggering: *Geophysical Research Letters*, v. 26, no. 24, p. 3693-3696.

Bernard, P., Briole, P., Meyer, B., Lyon-Caen, H., Gomez, J.-M., Tiberi, C., Berge, C., Cattin, R., Hatzfeld, D., and Lachet, C., 1997, The Ms= 6.2, June 15, 1995 Aigion earthquake (Greece):

evidence for low angle normal faulting in the Corinth rift: *Journal of Seismology*, v. 1, no. 2, p. 131-150.

We acknowledge and regret that our initial description and referencing of recorded low-angle normal fault earthquakes was incomplete. We now describe and cite the numerous M_w 6+ events in the final sentence of that paragraph, which reads:

“Furthermore, although earthquakes of up to M_w 6.8 have been reported on LANFs worldwide (e.g., Abers, 1991; Wernicke, 1995; Abers et al., 1997; Bernard et al., 1997; Axen, 1999; Abbott et al., 2001; as summarized in Collettini, 2011), multi-fault LANF ruptures may explain why M_w >7.0 earthquakes with well-resolved LANF nodal planes are absent in the modern instrumental record: seismic waveforms used in moment tensor inversions sample energy from all rupturing fault segments, and the contribution from LANF slip may be overprinted by simultaneous seismic slip on more steeply-dipping faults (Karlsson et al., 2021).”

4) Provide some comment as to why your models produce no footwall uplift (i.e. figure 3 and 4). Normal fault earthquakes such as the Borah Peak event in 1983 (Stein et al., 1988) and long-term geological extension produce substantial footwall uplift (Davis and Lister, 1988). Data from the footwall block of the Mai’iu fault indicate that footwall uplift is on the order of 5 – 10 km, many times larger than the hanging wall subsidence. A similar imbalance in footwall uplift relative to hanging wall subsidence is seen in the long term geological evolution of most low angle normal fault systems.

Davis, G. A., and Lister, G. S., 1988, Detachment faulting in continental extension; Perspectives from the southwestern U.S. Cordillera, in Clark, S. P., Burchfield, B. C., and Suppe, J., eds., *Processes in Continental Lithospheric Deformation*, Volume 218: Boulder, Geological Society of America, p. 133-160.

Stein, R. S., King, G. C. P., and Rundle, J. B., 1988, The growth of geological structures by repeated earthquakes 2. Field examples of continental dip-slip faults: *Journal of Geophysical Research*, v. 93, no. B11, p. 13,319-313,331.

We thank the reviewer for noting the interestingly minimal footwall uplift in these models. While we agree that limited coseismic footwall uplift seems out of line with longer-term uplift and subsidence patterns typical of LANFs, it may not be at odds with coseismic displacement patterns from other normal fault earthquakes. In the Borah Peak event, for example, the 11 cm of footwall uplift is relatively minor compared to the hanging-wall subsidence of 1.2 m. That earthquake occurred on a more steeply-dipping fault that would be expected to produce larger footwall uplift than a similar-magnitude rupture on a shallowly-dipping normal fault would, as illustrated nicely by this plot of relative vertical displacements predicted by elastic dislocation models for normal faults of various dips (Segall, 2010):

Figure 3.14. Effect of dip on the vertical component of displacement for a uniform-slip normal fault. Displacement normalized by slip.

Response Letter Figure 7. Plot of relative vertical displacements predicted by elastic dislocation models for slip on normal faults of various dips (Segall, 2010), highlighting the minimal footwall uplift expect for slip on a LANF dipping 30 degrees.

Our models generate up to 15 cm of footwall uplift (reference model without splays, Figure 3G), which may seem negligible, but is in fact of a similar order of magnitude as would be expected from 1-2 m of slip on such a shallowly dipping normal fault.

The compelling question of how to reconcile negligible coseismic footwall uplift with the long-term footwall uplift documented in many LANFs remains. One unfavorable possibility is that our models severely underrepresent coseismic footwall uplift; however, as discussed in the previous paragraph, static slip on such shallowly dipping normal faults is not expected to involve much more footwall uplift than that in our dynamic models. If LANF ruptures involve only minimal footwall uplift, the long-term footwall uplift recorded in many LANF systems must be predominantly accumulated during the interseismic and postseismic periods. This uplift could result from seismic-cycle fault-related processes like afterslip, interseismic creep, or postseismic viscoelastic relaxation. Alternatively, they could result from broader geodynamically driven gradual regional uplift (of both the footwall and hanging wall) coupled with localized coseismic displacements from LANF ruptures with minimal footwall uplift and pronounced hanging wall subsidence. We have now added a paragraph discussing footwall uplift to the section ‘Off-fault Damage in LANF Earthquakes’:

“As in simpler models (Biemiller et al, 2022; section 4.2), coseismic footwall uplift is notably minimal (<15 cm), seemingly at odds with longer-term patterns of footwall uplift documented in many LANF systems (e.g., Whitney et al., 2013) and recorded along the Mai’iu fault by exhumed fault rocks atop the ~3-km-tall Dayman-Suckling core complex (Mizera et al., 2019) and fossilized coral reefs emerged to >300 m elevation along the triangular-faceted coastline of Goodenough Bay (Mann & Taylor, 2002; Mann et al., 2009). If LANF ruptures involve pronounced hanging-wall subsidence with only minor footwall uplift, as observed in more

steeply-dipping normal fault earthquakes (e.g., Cheloni et al., 2017; Stein & Barrientos, 1985), then protracted LANF footwall uplift must accrue predominantly during the interseismic and/or postseismic periods, possibly via fault-related processes like interseismic creep, afterslip or viscoelastic relaxation, which may occur asymmetrically following large dip-slip earthquakes (e.g., Sun et al., 2014). Alternatively, broader geodynamic forcings insensitive to local fault locking could drive gradual regional uplift across both the footwall and hanging wall, upon which punctuated upper-crustal LANF earthquakes with minimal coseismic footwall uplift are superimposed, summing to a long-term net vertical displacement pattern with large footwall uplift and minor hanging-wall subsidence. Possible drivers of regional uplift in the highly extended settings where LANFs are commonly found include larger-scale geodynamic processes linked to long-lived localized rifting, such as isostatic compensation of warm, positively buoyant asthenospheric mantle material flowing into regions of thinned mantle lithosphere (e.g., Abers et al., 2002)."

Segall, P. (2010). *Earthquake and volcano deformation. Earthquake and Volcano Deformation.* Princeton University Press. <https://doi.org/10.5860/choice.48-0287>

Sun, T., Wang, K., Iinuma, T., Hino, R., He, J., Fujimoto, H., Kido, M., Osada, Y., Miura, S., Ohta, Y., & Hu, Y. (2014). Prevalence of viscoelastic relaxation after the 2011 Tohoku-oki earthquake. *Nature*, 514(7520), 84–87. <https://doi.org/10.1038/nature13778>

5) Minor point – I do not understand how the relative sea level fall revealed by the coral data are indicative of coseismic subsidence. In a subduction setting, coseismic subsidence is indicated by an abrupt sea level rise because the Earth's surface moves down relative to sea level. This is true for coastal estuaries (Atwater and Hemphill-Haley, 1997) and for coral heads above the Sumatra subduction zone (Zachariassen et al., 1999).

Atwater, B. F., and Hemphill-Haley, E., 1997, Recurrence intervals for great earthquakes of the past 3,500 years at northeastern Willapa Bay, Washington: USGS Profession Paper 1576, US Government Information Services [distributor], 92 p.

Zachariassen, J., Sieh, K., Taylor, F. W., Edwards, R. L., and Hantoro, W. S., 1999, Submergence and uplift associated with the giant 1833 Sumatran subduction earthquake; evidence from coral microatolls: *Journal of Geophysical Research*, v. 104, p. 895.

We appreciate this comment and agree that the coral data indicating episodic emergence seem most consistent with punctuated coseismic footwall uplift. In fact, the original coral study interpreted these data as evidence of episodic earthquakes involving pronounced footwall uplifts (Biemiller et al., 2020). However, subsequent dynamic rupture modeling of the Mai'iu fault along with further review of footwall uplift documented in other normal fault earthquakes have led us to reevaluate this interpretation. In our response to your comment #4 above (along with the associated paragraph added to the main text) and in Biemiller et al. (2022), we examine the possibility that documented deformation along the Mai'iu fault could result from episodic seismic slip involving minor footwall uplift and pronounced hanging-wall subsidence superimposed upon a geodynamically driven longer-wavelength gradual regional uplift signal. To explain the coral emergence data along Goodenough Bay, this process would need to

involve occasional or regular footwall capture of slices of the shallowest hanging wall related to the active slip surface in the shallowest crust stepping seaward onto outboard hanging wall splays. Though these combined processes may seem less intuitive than earthquake cycles with episodic large footwall uplifts, they appear equally if not more consistent with all the available data and models of this area.

Biemiller, J., Gabriel, A.-A., Ulrich, T., & Biemiller, J. (2022). The Dynamics of Unlikely Slip: 3D Modeling of Low-angle Normal Fault Rupture at the Mai'iu Fault, Papua New Guinea. *Geochemistry, Geophysics, Geosystems*, e2021GC010298.
<https://doi.org/10.1029/2021GC010298>

Biemiller, J., Taylor, F., Lavier, L., Yu, T.-L., Wallace, L., & Shen, C.-C. (2020). Emerged Coral Reefs Record Holocene Low-Angle Normal Fault Earthquakes. *Geophysical Research Letters*, 47(20).
<https://doi.org/10.1029/2020GL089301>

Clarity and context

The manuscript is clearly written. The context well articulated and the scope is reasonable.

References

See above comments for potential additional sources. Otherwise the references are appropriate and comprehensive.

My expertise

The focus of my research and teaching are structural geology, continental tectonics and geodynamics and earthquake geology.

Reviewer #3 (Remarks to the Author):

Review for “Dueling dynamics: competition between detachment rupture, splay faults, and off-fault damage.”

By J. Biemiller, A.-A. Gabriel, and T. Ulrich

The topic of this paper is very interesting and at the same time challenging one. The authors study the dynamic interaction between low angle normal faults and splay faults connected to the main structure. To do that they use both elastic and plastic models dynamic rupture models.

Low angle normal faults are by nature “enigmatic” with respect to the classical Andersonian theory for fault activation. Although we do not have any recent examples of major earthquakes where the main rupture occurs on a low angle normal fault, structures such as the Altotiberina fault in Italy could host damaging earthquakes. Same for the Mai'iu Papua New Guinea fault discussed in this paper.

The paper (and corresponding dynamic rupture finite element model) is based on a previously published work investigating the dynamics of shallow ruptures along the main detachment. The

new configuration includes six models each one corresponding to a different splay fault model with increasing dipping angles (45 to 75) and synthetic or antithetic geometry.

The family of models with synthetic (45 and 60) splay geometries shows two main and interesting observations when compared to the antithetic counterpart. First, rupture propagates easily over the splay fault, and second, the magnitude of slip is higher in the main detachment. In other words, rupture propagation on the splay fault, promotes higher slip in the main detachment. The antithetic configuration appears to have the opposite effect (act more like a barrier) and show lower slip on the main detachment.

In addition to the experiments with purely elastic rheology the authors presented a series of simulations with off-fault plastic failure, that allow them to reproduce observed subsidence detected using corals. The “plastic” experiments showed that when the sediments are weak and thick the model produces localized off-fault deformation matching historical subsidence. As expected, “strong” sediments do not show significant plastic yielding and for that reason generate a weaker and smoother subsidence pattern above the hanging wall.

This is a well written and nicely presented paper and I believe deserves publication after some clarifications/discussion regarding the results presented here. If you don't have space in the main text to provide comments/clarifications, you could extend the supplementary material:

We thank the reviewer for their time and effort reviewing this manuscript as well as for their comments and questions below, which we address individually.

1. In all the simulations presented in this paper the shallowest part of the main fault interface is velocity strengthening, inhibiting the propagation of rupture near the intersection with the free surface. We can appreciate this effect in the final slip map in the model with “no splay faults” (Figure 3G). Would that effect somehow “isolate” and promote rupture along the splay faults and basically create a preferential direction of rupture?

We first note that both the shallow parts of the main and splay faults in the original models are velocity-strengthening. How shallow frictional stability affects preferred rupture direction is an important question: we agree that the low rupture tendency of the shallowest LANF main fault (related to its gentle dip which is poorly oriented for slip and to the presence of velocity-strengthening fault materials in the shallow fault zone) likely promotes rupture propagating in a preferential direction more aligned with the steeply-dipping splays, particularly those that are synthetic to the main fault. However, given that all modeled faults (including the splays) have shallow velocity-strengthening properties, this preferential rupture direction more likely stems from mechanical effects of the LANF's shallow dip near the surface, such as its lower static slip tendency in an Andersonian stress field (e.g., Sibson, 1990; Collettini, 2011) and the stronger shallow dynamic clamping predicted for shallowly dipping normal faults (e.g., Nielsen, 1998; Oglesby et al., 1998; 2000; Aochi, 2018) compared with more steeply dipping normal faults.

To address this comment in the revised manuscript, we performed new models with velocity-weakening properties on both faults (Figure S11B) and on only the splay (Figure S11D), which show that stability of the splay fault has little effect on shallow rupture, while velocity-weakening materials in the main fault allow rupture to propagate slightly further up dip on it,

past the fault intersection at 5 km depth. These new results are shown in Figure S11, which is reproduced below our answer to the second half of the reviewer's comment.

Aochi, H. (2018). Dynamic asymmetry of normal and reverse faults due to constrained depth-dependent stress accumulation. *Geophysical Journal International*, 215(3), 2134–2143. <https://doi.org/10.1093/gji/ggy407>

Collettini, C. (2011). The mechanical paradox of low-angle normal faults: Current understanding and open questions. *Tectonophysics*, 510(3–4), 253–268. <https://doi.org/10.1016/j.tecto.2011.07.015>

Nielsen, S. B. (1998). Free surface effects on the propagation of dynamic rupture. *Geophysical Research Letters*, 25(1), 125–128. <https://doi.org/10.1029/97GL03445>

Oglesby, D. D., Archuleta, R. J., & Nielsen, S. B. (2000). The three-dimensional dynamics of dipping faults. *Bulletin of the Seismological Society of America*, 90(3), 616–628. <https://doi.org/10.1785/0119990113>

Oglesby, D. D., Archuleta, R. J., & Nielsen, S. B. (1998). Earthquakes on dipping faults: The effects of broken symmetry. *Science*, 280(5366), 1055–1059. <https://doi.org/10.1126/science.280.5366.1055>

Sibson, R. H. (1990). Rupture nucleation on unfavorably oriented faults. *Bulletin of the Seismological Society of America*, 80(6A), 1580–1604. <https://doi.org/10.1785/BSSA08006A1580>

In other words what would happen if rupture were allowed to propagate on both the splay fault and shallow part of the main detachment? I would expect for example that normal motion on the shallow detachment (after rupture passes the intersection with the splay and propagates towards the free surface) could potentially increase normal stress on the splay inhibiting rupture and consequently decrease the amount of final slip. When I first read the title of the paper, I thought that “Dueling” was referring to competing ruptures on the main detachment and the splay fault.

We first note that rupture was allowed to propagate onto both the splay fault and the shallow part of the main fault in the original models, as can be seen in Figure 3 C-F. We agree that shallow LANF rupture in those scenarios increases normal stress on the splay, clamping it and inhibiting further slip on it, which contributes to the lower total splay fault slip observed in these models (Figure 3 C-F). As the reviewer mentioned, ‘dueling’ refers to the competition between rupture of the main low-angle normal fault and the splay fault in these models, as well as the further competition with off-fault inelastic damage in the subsequent models with plasticity. To address this competition between splay rupture and LANF slip (which clamps the splays), we add a sentence to the second paragraph of the ‘Coseismic Reactivation of Pre-existing Splay Faults’ section of the revised manuscript:

“In contrast, LANF rupture propagates to the surface past the antithetic splays, inhibiting further splay slip by increasing normal stresses that clamp the splays.”

We believe that the reviewer is also asking how our results would change if the shallow detachment had fully velocity-weakening properties. In that case, we expect that the same general splay-geometry-dependent patterns of splay fault rupture and reduced detachment slip would arise (because the preferential rupture direction is not linked to the shallow velocity-weakening behavior, as discussed in the first part of this response), although shallow velocity-weakening detachment materials would likely allow rupture to propagate faster and further into the shallow portion of the detachment, leading to more slip over a larger area of the shallow detachment and slightly less slip on the splay. This scenario was previously explored in a model without splay faults or plasticity in Biemiller et al. (2022), Supporting Info Figure S5.

To address this question for the models of this study in the revised manuscript, we performed new dynamic rupture simulations with velocity-weakening properties on both faults (Figure S11B) and on only the splay (Figure S11D), which show that stability of the splay fault has little effect on shallow rupture, while velocity-weakening materials in the main fault allow rupture to propagate slightly further updip on it, past the fault intersection at 5 km depth. These new results are shown in Figure S11, which is reproduced below:

“Figure S11. Total slip after 15 s for A.) the model with a 60°-dipping synthetic splay fault (Figure 3B) and B-F.) additional variants of that model testing the effects of variable fault friction and stress. B.) Model with shallow velocity-weakening friction on both faults results in more total slip and allows rupture to penetrate slightly further updip on the main fault, above the fault intersection at 5 km depth, highlighting the stabilizing effects of clay-rich velocity-strengthening gouges in these fault systems. C-E.) Models with friction and stress conditions

designed to test the mechanical influence of variable levels of fault maturity between the long-lived, high-offset Mai'iu fault and the younger, immature low-offset splay. C.) A weaker main fault with lower effective static friction decreasing above 15 km depth to $f_0 = 0.3$ at the surface results in similar but muted additional slip than in B, suggesting that the strength of the main fault may be less important than its frictional stability in modulating shallow rupture patterns. D.) Model with identical conditions to C except velocity-weakening materials in the splay fault generates indistinguishable slip patterns from those of C, suggesting splay fault slip is largely insensitive to splay fault frictional stability. E.) Model with lower initial stresses calculated based on a weaker coefficient of static friction of 0.45 results in similar patterns of main and splay fault slip but with significantly less total slip, while F.) one with even weaker stresses and static friction of 0.3 fails to nucleate sustained dynamic rupture. E & F suggest that if fault strength were governed by the weak velocity-strengthening gouges in the mature LANF core, interseismic loading would be relieved by creep and thus interseismic stresses could not accumulate to levels capable of producing the large paleoearthquakes recorded in these LANF systems."

Biemiller, J., Gabriel, A.-A., Ulrich, T., & Biemiller, J. (2022). The Dynamics of Unlikely Slip: 3D Modeling of Low-angle Normal Fault Rupture at the Mai'iu Fault, Papua New Guinea. *Geochemistry, Geophysics, Geosystems*, e2021GC010298. <https://doi.org/10.1029/2021GC010298>

To clarify the mechanics favoring splay fault rupture over shallow detachment rupture, we add the following sentence to the first "Discussion & Conclusion" paragraph:

"Furthermore, synthetic splays are better-oriented for slip than the shallow detachment in both the static and dynamic sense: steeper normal faults embedded in Byerlee materials and dipping up to 65-70° are preferentially oriented for slip under Andersonian extensional stresses (e.g., Sibson, 1990; Abers, 2009; Collettini, 2011), while free surface stress interactions with an updip-propagating normal fault rupture promotes further rupture propagation on faults dipping 30-75° but inhibits it for those dipping <30° (Oglesby et al., 1998)."

2. Have you looked specifically at the normal stress changes ahead and behind the rupture front and how such changes may affect the outcome of your simulations? Would rupture along a normal fault produce clamping or unclamping ahead of the rupture front? How such process affects the interaction with the splay fault?

In the presence of a free surface, slip on portions of a rupturing normal fault behind the rupture front generally tends to increase the normal stress and resistive frictional force on fault patches ahead of the rupture, clamping these areas and inhibiting further rupture propagation (Oglesby et al., 1998; Aochi, 2018). However, this clamping is mitigated at shallow depths by stress interactions with the free surface, which weaken and promote rupture propagation into the shallow portion of the fault for normal faults dipping 30-75° (Oglesby et al., 1998; Nielsen, 1998). Thus, free surface interactions aid rupture on the steeply dipping splay faults but

inhibits rupture of the shallow detachment, which in conjunction with the clamping ahead of the rupture front may quite efficiently arrest updip rupture of the shallow detachment, further promoting shallow slip on the steeper splays and enhancing the preferential rupture direction discussed in our response to comment 1. To confirm that these previously documented clamping effects occur in our models, we plot the maximum transient normal stress near the hypocenter after 2 seconds of the reference model without splay faults or plasticity (Figure 3G) along with white 0.5 s rupture contours (mapview of the fault; updip towards the lower left):

SeisSol's conventions use negative normal stress in compression. Thus, the strong negative transient stress bands (purple) ahead of the rupture front indicate clamping ahead of the updip-propagating rupture induced by slip on the deeper ruptured part of the fault, as expected based on previous normal fault dynamic rupture modeling studies. In contrast to this clamping, the synthetic splay fault in the model in Figure 3B experiences lower normal stresses and dynamic unclamping ahead of the updip-propagating rupture, as illustrated by these plots of maximum transient normal stress and rupture contours from that model:

With 60°-dipping synthetic splay fault:

This unclamping helps rupture jump ahead onto the splay fault before the main rupture front reaches the intersection of the splay and the low-angle normal fault, as shown in the right panel above. To illustrate these interesting clamping and unclamping effects in the revised manuscript, we now include these plots as Supplementary Figure S6.

Oglesby, D. D., Archuleta, R. J., & Nielsen, S. B. (1998). Earthquakes on dipping faults: The effects of broken symmetry. *Science*, 280(5366), 1055–1059. <https://doi.org/10.1126/science.280.5366.1055>

Aochi, H. (2018). Dynamic asymmetry of normal and reverse faults due to constrained depth-dependent stress accumulation. *Geophysical Journal International*, 215(3), 2134–2143. <https://doi.org/10.1093/gji/ggy407>

Nielsen, S. B. (1998). Free surface effects on the propagation of dynamic rupture. *Geophysical Research Letters*, 25(1), 125–128. <https://doi.org/10.1029/97GL03445>

3. Does the speed of rupture on the main detachment (slow vs fast rupture) plays any role on how efficiently the system activates the splay faults?

We thank the reviewer for this interesting question. Rupture velocity likely determines whether the competition between detachment slip, splay faulting, and off-fault damage is mechanically dominated by static or dynamic processes. That is, slow ruptures with less energy at the rupture front induce weaker dynamic stress perturbations than their fast, energetic counterparts. Whether slow rupture or slow slip propagates onto the shallow detachment or slips the splay fault would thus be more sensitive to how favorably or unfavorably oriented for slip each fault is relative to the far-field tectonic stress field, as predicted by static Andersonian-type fault mechanics (e.g., Sibson, 1990). Fast ruptures, on the other hand, energetically and rapidly propagate updip and along-strike, transmitting strong dynamic stresses that interact with the free surface as well as neighboring faults. Thus, preferential rupture of the LANF or

splay faults would be more strongly influenced by dynamic processes such as interaction with the free surface for fast ruptures than for their slow-rupturing counterparts.

Rupture velocity mirrors the available rupture energy, with energetic fast ruptures propagating past structural or rheological slip barriers much more easily than slow ruptures. Processes dissipating rupture energy must act more efficiently in order to fully arrest a fast strong rupture. In our models, rupture velocity is one indicator of how much energy is available for splay fault slip by the time rupture reaches the splay-detachment intersection. To illustrate how rupture energy affects splay-detachment dynamics, we compare plots of total slip and 0.5 s rupture contours from the model with a 60°-dipping synthetic splay fault (Figure 3B) and one with the same geometry and parameters but with higher pore fluid pressure ($\lambda_f = 0.66$) and weaker nucleation stresses (up to 30 MPa) applied over a smaller hypocentral area of radius 2 km.

Right: model with 60°-dipping splay fault from Figure 3B
Left: identical model except with higher pore pressure and a weaker nucleation

Rupture immediately propagates faster and involves higher slip in the model with more available rupture energy (right column). Rupture jumps to the splay fault earlier in this model than in that with the slower rupture. The fast rupture is so energetic when it reaches the splay-detachment intersection and free surface that a strong back-propagating reflected phase is transmitted back to the deeper detachment, establishing a secondary rupture front that subsequently tails the main laterally propagating rupture. This reflected phase generates the high secondary slip accrued on the deeper detachment from $t = 10 - 14$ s. The reflected rupture can be better tracked in plots of the instantaneous slip rates, shown below:

Right: model with 60°-dipping splay fault from Figure 3B

Left: identical model except with higher pore pressure and a weaker nucleation

Overall, rupture velocity depends on the balance between fracture energy and energy release rate (e.g., Weng & Ampuero, 2022) which strongly influences total slip, total rupture extent, and whether rupture arrests prematurely when encountering rupture barriers. To highlight the role of rupture energy in these models, we add these plots as new Supplementary Figures S7 and S8.

Sibson, R. H. (1990). Rupture nucleation on unfavorably oriented faults. *Bulletin of the Seismological Society of America*, 80(6A), 1580–1604.
<https://doi.org/10.1785/BSSA08006A1580>

Weng, H., & Ampuero, J.-P. (2022). Integrated rupture mechanics for slow slip events and earthquakes. *Nature Communications*, 13(1), 7327. <https://doi.org/10.1038/s41467-022-34927-w>

We address these issues in the revised manuscript by including these plots in Figures S7 and S8 and adding this discussion of rupture velocity in the caption of Figure S7:

Right: model with 60°-dipping splay fault from Figure 3B
 Left: identical model except with higher pore pressure and a weaker nucleation

Figure S7. Right column: Evolution of total slip with 0.5 s rupture contours for the model with a 60°-dipping synthetic splay fault (Figure 3B); Left column: Evolution of total slip with 0.5 s rupture contours for a model with identical geometry and parameters except with higher pore fluid pressure ($\lambda_f = 0.66$) and weaker nucleation stresses (up to 30 MPa) applied over a smaller

hypocentral area of radius 2 km. These models illustrate how initial stress conditions affect resulting rupture velocities, and the balance between fracture energy and energy release rate (e.g., Weng & Ampuero, 2022) relate to modeled splay-detachment dynamics. Slow ruptures with less energy at the rupture front induce weaker dynamic stress perturbations than their fast, energetic counterparts. Whether slow rupture or slow slip propagates onto the shallow detachment or slips the splay fault would thus be more sensitive to how favorably or unfavorably oriented for slip each fault is relative to the far-field tectonic stress field, as predicted by static Andersonian-type fault mechanics (e.g., Sibson, 1990). Fast ruptures, on the other hand, energetically and rapidly propagate updip and along-strike, transmitting strong dynamic stresses that interact with the free surface as well as neighboring faults. Thus, preferential rupture of the detachment or splay faults would be more strongly influenced by dynamic processes such as rupture interaction with the free surface for fast ruptures than for their slower-rupturing counterparts. Rupture velocity mirrors the available rupture energy, with energetic fast ruptures propagating past structural or rheological slip barriers much more easily than slower ruptures. Processes dissipating rupture energy must act more effectively in order to fully arrest a fast strong rupture.

Right: model with 60°-dipping splay fault from Figure 3B
 Left: identical model except with higher pore pressure and a weaker nucleation

Figure S8. Right column: Dip-slip rate for the model with a 60°-dipping synthetic splay fault (Figure 3B); Left column: Dip-slip rate for a model with identical geometry and parameters

except with higher pore fluid pressure ($\lambda_f = 0.66$) and weaker nucleation stresses (up to 30 MPa) applied over a smaller hypocentral area of radius 2 km.”

4. Could a rupture start from a splay fault instead of the main detachment?

Good question. While it's feasible that rupture could nucleate on a splay fault and propagate onto or trigger rupture of the main detachment resulting in a large earthquake, this scenario seems less likely than nucleation on the detachment. First, compelling evidence from slip distributions and stress inversions of the 2010 Mw 7.2 El Mayor-Cucapah earthquake indicate that well-oriented faults in that system were critically stressed well before the rupture which nucleated on an underlying, shallowly dipping detachment fault (Fletcher et al., 2016). These observations led those authors to propose that shallowly dipping detachments act as keystone faults in extensional systems, inhibiting rupture of neighboring and intersecting faults until the detachment is critically stressed and can nucleate rupture of a large multi-fault event.

More general mechanical considerations also suggest nucleation on splay faults is unlikely. First, slip initiating on a splay would increase the normal stress and clamp portions of the detachment: synthetic splay fault slip would clamp the deeper detachment, inhibiting rupture downdip; antithetic splay fault slip would clamp the shallower detachment, inhibiting rupture to the surface but potentially unclamping the deeper detachment and promoting secondary nucleation there. In addition, near the splay-detachment intersection, splay slip could relieve some of the shear stress previously accumulated on the detachment, pushing it further from failure. Overall, though large ruptures nucleating on a splay fault cannot be ruled out entirely, splay faults in active LANF detachment systems seem more prone to interseismic creep and microseismic brittle failure or repeating earthquakes, as observed on splays of the Altotiberina fault (Valoroso et al., 2017; Vuan et al., 2020) and inferred from the El Mayor-Cucapah event (Fletcher et al., 2016).

To address the possibility of large ruptures nucleating on splay faults in the main text, we add this sentence to the first paragraph of the 'Discussion & Conclusions' section:

“Despite their favorable orientations, critically stressed splay faults appear unlikely to nucleate large ruptures, slipping instead via bursts of microseismic creep and small earthquake (e.g., Valoroso et al., 2017; Vuan et al., 2020) during the interseismic period of the underlying LANF, which may act as a 'keystone fault' that prevents splays from slipping unstably between large LANF ruptures (Fletcher et al., 2016).”

Fletcher, J. M., Oskin, M. E., & Teran, O. J. (2016). The role of a keystone fault in triggering the complex El Mayor-Cucapah earthquake rupture. *Nature Geoscience*, 9(4), 303–307. <https://doi.org/10.1038/ngeo2660>

Valoroso, L., Chiaraluce, L., Di Stefano, R., & Monachesi, G. (2017). Mixed-Mode Slip Behavior of the Altotiberina Low-Angle Normal Fault System (Northern Apennines, Italy) through High-

Resolution Earthquake Locations and Repeating Events. *Journal of Geophysical Research: Solid Earth*, 122(12), 10,220-10,240. <https://doi.org/10.1002/2017JB014607>

Vuan, A., Brondi, P., Sukan, M., Chiaraluce, L., di Stefano, R., Michele, M. (2020). Intermittent Slip Along the Alto Tiberina Low-Angle Normal Fault in Central Italy. *Geophysical Research Letters*, 47(17), 1–11. <https://doi.org/10.1029/2020GL089039>

5. A short discussion on what could be some limiting factors in your work. For example, would a realistic surface topography in your FEM make any difference in your simulations?

We appreciate the suggestion to acknowledge possible limiting factors in our work. Realistic surface topography would slightly affect shallow rupture characteristics in these models, but it is unlikely to have any major effect that would alter the large-scale deformation patterns and mechanical relationships outlined in this study. The most distinct topographic features in this region are the ~3 km tall domal mountains in the footwall of the Mai'iu fault. In contrast, the topography and bathymetry of the hanging wall is characterized by a smooth gently seaward-dipping surface. Previous dynamic rupture models of variably-dipping normal faults showed that dynamic stress interactions between the rupture front and the free surface exert a strong effect on effective stresses on the shallow portion of the fault and the extent of updip rupture propagation (Nielsen, 1998; Oglesby et al., 1998, 2000; Aochi, 2018). Rupture interacts most strongly with the nearest portions of the free surface, which in the case of a normal fault are those above the hanging wall. Thus, despite the typical pronounced footwall topography of major LANFs, the strongest free surface effects on rupture propagation should stem from the relatively flat hanging wall which is not dissimilar from the flat topography used here.

The largest limitation of our models is very likely the absence of seismic and geodetic observations from a modern large Mai'iu fault earthquake or even an analogous large low-angle normal fault rupture. Although we believe the geologic and geophysical observations from this region provide enough constraints to construct meaningful and informative dynamic rupture simulations, they nonetheless lack the constraints on instrumentally observable rupture characteristics that help inform and tightly constrain dynamic rupture studies of well-documented modern earthquakes. The scarcity and uncertainty of coseismic data was originally discussed in the last paragraph of the section "Off-fault damage in detachment earthquakes." To further address how data limitations translate to model limitations, we add a sentence to this paragraph which reads:

"Nonetheless, we acknowledge that relative to dynamic rupture modeling studies of well-documented earthquakes, our models are limited by the absence of seismological data from a modern analogue earthquake, leading to fewer constraints on and higher uncertainties in instrumentally observable rupture characteristics like hypocentral location, stress drop, and rupture velocity."

Aochi, H. (2018). Dynamic asymmetry of normal and reverse faults due to constrained depth-dependent stress accumulation. *Geophysical Journal International*, 215(3), 2134–2143. <https://doi.org/10.1093/gji/ggy407>

Nielsen, S. B. (1998). Free surface effects on the propagation of dynamic rupture. *Geophysical Research Letters*, 25(1), 125–128. <https://doi.org/10.1029/97GL03445>

Oglesby, D. D., Archuleta, R. J., & Nielsen, S. B. (2000). The three-dimensional dynamics of dipping faults. *Bulletin of the Seismological Society of America*, 90(3), 616–628. <https://doi.org/10.1785/0119990113>

Oglesby, D. D., Archuleta, R. J., & Nielsen, S. B. (1998). Earthquakes on dipping faults: The effects of broken symmetry. *Science*, 280(5366), 1055–1059. <https://doi.org/10.1126/science.280.5366.1055>

REVIEWER COMMENTS

Reviewer #1 (Remarks to the Author):

As previously mentioned, this is a well written paper focused on investigating the possible effects to dynamic rupture simulation models of an active low-angle normal fault of adding further complications to an already complex 3D rupture model. The authors investigate the parameter space of adding a more steeply dipping splay of variable dip (both synthetic and antithetic) and the possible effects of shallow off-fault sediment deformation of various strength and thickness in the hanging wall. These results are especially of interest to those concerned with the degree to which adding such elements of increased fault or off-fault complexity can potentially enhance our understanding of how dynamic rupture deformation may be expressed in the shallow geologic record.

The authors do a very commendable job of addressing reviewers' comments and concerns, and as a result, the revised paper is now --I believe-- much improved and well worth publishing. My compliments.

There are, however, a few minor remaining issues I have regarding certain statements that can be easily addressed with minor text revision. These issues concern: 1) lingering misleading statements between the reference model used and the previously published one (Biemiller et al., 2022); and 2) lingering misleading statements regarding how the complex 3D fault geometry is generally characterized and defined.

1) The original manuscript text and the new revised manuscript both indicate that: (Line 137) *We first consider six models subject to Andersonian extension based on the preferred LANF-only earthquake model of Biemiller et al (2022)*. Biemiller et al. (2022) state: *...we describe four key LANF dynamic rupture models including one preferred reference model.*

Taken together, these two statements imply that the rupture models in the new study are based on the 'one preferred reference model' of Biemiller et al. (2022), which is in fact not true. The preferred reference model from 2022 has a maximum slip at depth of about 3-4 m and a moment magnitude of Mw 7.1. The reference model used in this paper has a maximum slip of about 7 m and a Mw 7.4. The authors explained that they did indeed change the initial conditions of the 'preferred' reference model and why they changed it, not realizing that all I was asking is that they clearly state that the reference model was changed and how it was changed. Although they did explain how it was changed, they left unchanged the original, misleading text in the main body (Line 137) that created the confusion in the first place. Thus, all that was needed, at least from my perspective, was to simply modify the text of Line 137 to read something like:

We first consider six models subject to Andersonian extension based on and similar to the preferred LANF-only earthquake model of Biemiller et al (2022), however, the models in this study use a wider nucleation region combined with stronger overstress and lower, near-hydrostatic pore fluid pressures. This results in a reference model with generally larger maximum slip at depth near the hypocenter (7 m) and larger moment release (Mw 7.4).

2) Based on Line 103-105 [*We perform and analyze physics-based 3D dynamic rupture simulations of the active concave-down Mai'iu fault in Papua New Guinea (Fig. 2A)*], and the relatively simple, concave-down geometry shown in Fig.2A and in Biemiller et al. (2022, Fig.3), I presumed the modeled fault surface had this same simple curved shape. This simple shape appeared to be inconsistent with the more complex fault geometry shown in Figs.2E & 2F. The fault geometry shown in Figs. 2E & 2F has variable deep dip along strike of 40°, 43°, and a planar dip of 20° to 15 km depth along its eastern edge. This did not match the stated geometry of 'concave-downward' and being 'identical to that described in Biemiller et al. (2022, Fig.3)', or with the resolvable dip at depth defined by seismicity.

The downdip extent of the fault --based on microseismicity (Eilon et al., 2015; Abers et al., 2016)-- is only able to define a rather simple, planar geometry at depth with relatively constant dip of about 30°. The seismicity does not define a planar surface that

dips at 20°, nor does it provide sufficient resolution to define a complex surface at depth with variable steeper dip ($\geq 40^\circ$) as modeled by Webber et al. (2020), and shown in Figs. 2E, 2F & S10. I understand that this is not an issue for the authors, as they use the 3D surface previously defined and published by Webber et al. (2020). I just thought they could be a bit more circumspect about claiming the modeled surface is everywhere 'concave-downward', or is 'identical to that shown in Biemiller et al. (2020, Fig.3)', when it is in fact not always the case. Again, this just may need minor revision of the text (e.g., 'mostly concave-downward' rather than 'concave-downward') to avoid some unnecessary confusion.

We thank the reviewer for carefully evaluating our revised manuscript and providing constructive suggestions for improvement in their reviews. We appreciate the opportunity to address the remaining issues and further clarify aspects of our models in the further revised manuscript attached here. Below, we respond to reviewer 1's comments and detail the changes made in the revised manuscript to address these issues.

Reviewer #1 (Remarks to the Author):

As previously mentioned, this is a well written paper focused on investigating the possible effects to dynamic rupture simulation models of an active low-angle normal fault of adding further complications to an already complex 3D rupture model. The authors investigate the parameter space of adding a more steeply dipping splay of variable dip (both synthetic and antithetic) and the possible effects of shallow off-fault sediment deformation of various strength and thickness in the hanging wall. These results are especially of interest to those concerned with the degree to which adding such elements of increased fault or off-fault complexity can potentially enhance our understanding of how dynamic rupture deformation may be expressed in the shallow geologic record.

The authors do a very commendable job of addressing reviewers' comments and concerns, and as a result, the revised paper is now --I believe-- much improved and well worth publishing. My compliments. There are, however, a few minor remaining issues I have regarding certain statements that can be easily addressed with minor text revision. These issues concern: 1) lingering misleading statements between the reference model used and the previously published one (Biemiller et al., 2022); and 2) lingering misleading statements regarding how the complex 3D fault geometry is generally characterized and defined.

1) The original manuscript text and the new revised manuscript both indicate that: (Line 137) *We first consider six models subject to Andersonian extension based on the preferred LANF-only earthquake model of Biemiller et al (2022).* Biemiller et al. (2022) state: *...we describe four key LANF dynamic rupture models including one preferred reference model.*

Taken together, these two statements imply that the rupture models in the new study are based on the 'one preferred reference model' of Biemiller et al. (2022), which is in fact not true. The preferred reference model from 2022 has a maximum slip at depth of about 3-4 m and a moment magnitude of Mw 7.1. The reference model used in this paper has a maximum slip of about 7 m and a Mw 7.4. The authors explained that they did indeed change the initial conditions of the 'preferred' reference model and why they changed it, not realizing that all I was asking is that they clearly state that the reference model was changed and how it was changed. Although they did explain how it was changed, they left unchanged the original, misleading text in the main body (Line 137) that created the confusion in the first place. Thus, all that was needed, at least from my perspective, was to simply modify the text of Line 137 to read something like:

We first consider six models subject to Andersonian extension based on and similar to the preferred LANF-only earthquake model of Biemiller et al (2022), however, the models in this study use a wider nucleation region combined with stronger overstress and lower, near-hydrostatic pore fluid pressures. This results in a reference model with generally larger maximum slip at depth near the hypocenter (7 m) and larger moment release (Mw 7.4).

We thank the reviewer for pointing out this oversight and regret not clarifying that introductory paragraph in the original revisions. We also appreciate their suggestion for how to address this issue, which we have incorporated into the revised version that now reads:

“We first consider six models subject to Andersonian extension similar to the preferred LANF-only fully elastic earthquake model of Biemiller et al (2022) (Fig. 2A; see Supp. Text S1 for details), but with a wider nucleation region combined with stronger overstress and lower, near-hydrostatic pore fluid pressure to balance the effects of off-fault plasticity while yielding comparable earthquake scenarios. The adapted reference model with larger maximum slip at depth near the hypocenter (7 m) results in an overall slightly larger moment release (M_w 7.4).”

2) Based on Line 103-105 [We perform and analyze physics-based 3D dynamic rupture simulations of the active concave-down Mai'iu fault in Papua New Guinea (Fig. 2A)], and the relatively simple, concave-down geometry shown in Fig.2A and in Biemiller et al. (2022, Fig.3), I presumed the modeled fault surface had this same simple curved shape. This simple shape appeared to be inconsistent with the more complex fault geometry shown in Figs.2E & 2F. The fault geometry shown in Figs. 2E & 2F has variable deep dip along strike of 40°, 43°, and a planar dip of 20° to 15 km depth along its eastern edge. This did not match the stated geometry of 'concave-downward' and being 'identical to that described in Biemiller et al. (2022, Fig.3)', or with the resolvable dip at depth defined by seismicity. The downdip extent of the fault --based on microseismicity (Eilon et al., 2015; Abers et al., 2016)-- is only able to define a rather simple, planar geometry at depth with relatively constant dip of about 30°. The seismicity does not define a planar surface that dips at 20°, nor does it provide sufficient resolution to define a complex surface at depth with variable steeper dip ($\geq 40^\circ$) as modeled by Webber et al. (2020), and shown in Figs. 2E, 2F & S10. I understand that this is not an issue for the authors, as they use the 3D surface previously defined and published by Webber et al. (2020). I just thought they could be a bit more circumspect about claiming the modeled surface is everywhere 'concave-downward', or is 'identical to that shown in Biemiller et al. (2020, Fig.3)', when it is in fact not always the case. Again, this just may need minor revision of the text (e.g., 'mostly concave-downward' rather than 'concave-downward') to avoid some unnecessary confusion.

We appreciate this comment and regret not adequately addressing the reviewer's initial concerns about the fault geometry in the original revisions. There appear to be a few issues behind the lingering questions about the modeled fault geometry, which we address and hopefully clarify below.

First, we acknowledge and fully agree that our original description of the fault geometry as simply 'concave-down' may have been misleading or unclear. The segment exhuming the Dayman Dome appears to be concave-down, but the entire fault surface is strongly megacorrugated along-strike. For example, the section below the synclinal Gwoira rider block is locally concave-up. Interestingly, this folding appears to occur synexhumationally, such that the corrugations are amplified near the surface but more muted at depth. We have now revised that sentence to describe the fault geometry more accurately:

“We perform and analyze physics-based 3D dynamic rupture simulations of the active, mega-corrugated and predominantly concave-down Mai'iu fault in Papua New Guinea (Fig. 2A),”

Second, we acknowledge the inherent uncertainties in our modeled fault geometry. There are very few (if any) faults for which we precisely know their subsurface geometry. In the absence of high-resolution seismic imaging or abundant, evenly distributed microseismicity, modelers must make some assumptions and infer fault geometries based on the available data. For the Mai'iu fault, the constraints on fault geometry at depth come primarily from field mapping (not only of the surface dip and dips of exhumed fault remnants, but also of the footwall geomorphology, which reflects large-scale fault geometry at depth, e.g. the megacorrugations) and sparse microseismicity. Webber et al. (2020) used common structural modeling techniques to construct their preferred fault surface geometry which attempts to match field-mapped dips and deeper microseismicity:

There is still plenty of uncertainty in their modeled fault geometry, which is amplified farther away from the sources of input data constraints. To ensure these uncertainties are appropriately acknowledged in our revised manuscript, we have added the following sentence to the caption of Figure 1:

“Given that Webber et al. (2020)’s surface is largely constrained by onshore mapping along the Dayman-Gwoira segments and microseismicity below 15 km depth, the modeled fault geometry is less tightly constrained and has larger uncertainties towards the along-strike edges and deeper portions of the fault.”

Finally, the reviewer notes an apparent discrepancy between the fault geometry used in our simulations and that used in Biemiller et al. (2022). This is not the case; we use the same fault geometry in both studies. We assume this apparent discrepancy may stem from the clipped view of the fault shown in Figures 2 E & F, which was not explicitly mentioned in the text and was perhaps difficult to distinguish from the figure alone. In many plots throughout the manuscript, we show only the upper 15 km of the model, as the strong velocity-strengthening behavior and low deviatoric stresses below this depth prevents rupture from ever propagating deeper than 15 km in our models. While the fault geometry model we adopted from Webber et al. (2020) appears rectangular in mapview, it extends deeper below some segments than below others, leading to the “triangular” downdip geometry seen in Figure 2A (and in Fig. 3A from Biemiller et al., 2022). Although this triangular downdip portion is included in the numerical simulations, it is effectively inert due to its low stress and stable friction. Thus, we include plots of only the upper 15 km of the models, where dynamic rupture occurs. We now better explain that the Mai’iu fault geometry used here is the same geometry used in Biemiller et al. (2022), where it was more thoroughly described in section 2.3.1 as:

“Our non-planar fault geometry (Figure 3a,b) is based on that of Webber et al. (2020), who combined surface dip measurements with the microseismic data of Abers et al. (2016) to constrain an interpolated subsurface model of the distinctively corrugated Mai’iu fault. Near the base of the rapidly exhuming Mt. Dayman, the active fault geometry mirrors the concave-down morphology of the domal footwall and exhumed fault surface of Mizera et al. (2019). Along-strike to the southeast, the fault shallows and becomes concave-up beneath the Gwoira rider block, a slice of the original hanging wall captured within the footwall of the active Gwoira splay fault dipping 37-44°. Further southeast, the Mai’iu fault steps offshore where seismic reflection data indicate pervasive normal faulting of its hanging wall basin (Fitz, 2011; Fitz & Mann, 2013a, 2013b), although the lack of surface exposure reduces the accuracy of the fault model here.”

To clarify this apparent discrepancy in the revised manuscript, we highlight that only the upper 15 km of the model is shown in certain plots by editing a sentence in Figure 2’s caption to read:

“Note that although E & F show only the upper 15 km of the model, the full modeled fault geometry is identical to that of Biemiller et al. (2022) (Section 2.3.1; Fig. 3), which was derived from Webber et al. (2020).”

We edit part of Figure 3’s caption to read:

“A-F.) Fault slip above 15 km depth (top: map view; inset: oblique view along-strike)”

We edit part of Figure 4’s caption to read:

“A-F.) Fault slip above 15 km depth, clipped plastic strain (top: map view; inset: oblique view along-strike), and surface uplift after 15 seconds in models accounting for off-fault inelastic failure with shallow sedimentary basins of variable strength and thickness.”

We edit part of Figure 5’s caption to read:

“A-B.) Fault slip above 15 km depth and clipped plastic strain after 15 seconds in models with plasticity, thick intermediate-strength sediments, and pre-existing splay faults dipping 45° antithetic (A) or

synthetic (B) to the LANF, highlighting the dynamic competition between shallow LANF slip, splay fault slip, and off-fault plastic failure.”